# Revealing architectural order with quantitative label-free imaging and deep learning

Syuan-Ming Guo[1†], Li-Hao Yeh[1†], Jenny Folkesson[1†], Ivan E Ivanov[1‡], Anitha P Krishnan[1‡§], Matthew G Keefe[2‡], Ezzat Hashemi[3], David Shin[2], Bryant B Chhun[1], Nathan H Cho[1#], Manuel D Leonetti[1], May H Han[3], Tomasz J Nowakowski[2], Shalin B Mehta[1*]

[1]Chan Zuckerberg Biohub, San Francisco, United States; [2]Department of Anatomy, University of California, San Francisco, San Francisco, United States; [3]Department of Neurology, Stanford University, Stanford, United States

**Abstract** We report quantitative label-free imaging with phase and polarization (QLIPP) for simultaneous measurement of density, anisotropy, and orientation of structures in unlabeled live cells and tissue slices. We combine QLIPP with deep neural networks to predict fluorescence images of diverse cell and tissue structures. QLIPP images reveal anatomical regions and axon tract orientation in prenatal human brain tissue sections that are not visible using brightfield imaging. We report a variant of U-Net architecture, multi-channel 2.5D U-Net, for computationally efficient prediction of fluorescence images in three dimensions and over large fields of view. Further, we develop data normalization methods for accurate prediction of myelin distribution over large brain regions. We show that experimental defects in labeling the human tissue can be rescued with quantitative label-free imaging and neural network model. We anticipate that the proposed method will enable new studies of architectural order at spatial scales ranging from organelles to tissue.

**\*For correspondence:**
shalin.mehta@czbiohub.org

[†]These authors contributed equally to this work
[‡]These authors also contributed equally to this work

**Present address:** [§]Genentech, San Francisco, United States; [#]University of California, San Francisco, San Francisco, United States

**Competing interests:** The authors declare that no competing interests exist.

## Introduction

The function of living systems emerges from the interaction of its components over spatial and temporal scales that range many orders of magnitude. Light microscopy is uniquely useful to record dynamic arrangement of molecules within the context of organelles, of organelles within the context of cells, and of cells within the context of tissues. Combination of fluorescence imaging and automated analysis of image content with deep learning (*Moen et al., 2019*; *Belthangady and Royer, 2019*; *Van Valen et al., 2016*) has opened new avenues for understanding complex biological processes. However, characterizing the architecture and dynamics with fluorescence remains challenging in many important biological systems. The choice of label can introduce observation bias in the experiment and may perturb the biological process being studied. For example, labeling cytoskeletal polymers often perturbs their native assembly kinetics (*Belin et al., 2014*). Genetic labeling of human tissue and non-model organisms is not straightforward and the labeling efficiency is often low. Labeling with antibodies or dyes can lead to artifacts and requires careful optimization of the labeling protocols. The difficulty of labeling impedes biological discoveries using these systems. By contrast, label-free imaging requires minimal sample preparation as it measures the sample's intrinsic properties. Lable-free imaging is capable of visualizing many biological structures simultaneously with minimal photo-toxicity and no photo-bleaching, making it particularly suitable for live-cell imaging. Measurements made without label are often more robust since experimental errors associated with the labeling are avoided. Multiplexed imaging with fluorescence and label-free contrasts

**eLife digest** Microscopy is central to biological research and has enabled scientist to study the structure and dynamics of cells and their components within. Often, fluorescent dyes or trackers are used that can be detected under the microscope. However, this procedure can sometimes interfere with the biological processes being studied.

Now, Guo, Yeh, Folkesson et al. have developed a new approach to examine structures within tissues and cells without the need for a fluorescent label. The technique, called QLIPP, uses the phase and polarization of the light passing through the sample to get information about its makeup.

A computational model was used to decode the characteristics of the light and to provide information about the density and orientation of molecules in live cells and brain tissue samples of mice and human. This way, Guo et al. were able to reveal details that conventional microscopy would have missed. Then, a type of machine learning, known as 'deep learning', was used to translate the density and orientation images into fluorescence images, which enabled the researchers to predict specific structures in human brain tissue sections.

QLIPP can be added as a module to a microscope and its software is available open source. Guo et al. hope that this approach can be used across many fields of biology, for example, to map the connectivity of nerve cells in the human brain or to identify how cells respond to infection. However, further work in automating other aspects, such as sample preparation and analysis, will be needed to realize the full benefits.

---

enables characterization of the dynamics of labeled molecules in the context of organelles or cells. Thus, label-free imaging provides measurements complementary to fluorescence imaging for a broad range of biological studies, from analyzing architecture of archival human tissue to characterizing organelle dynamics in live cells.

Classical label-free microscopy techniques such as phase contrast (*Zernike, 1955*), differential interference contrast (DIC) (*Nomarski, 1955*), and polarized light microscopy (*Schmidt, 1926*; *Inoue, 1953*) are qualitative. They turn specimen-induced changes in phase (shape of the wavefront) and polarization (the plane of oscillation of the electric field) of light into intensity modulations that are detectable by a camera. These intensity modulations are related to specimens' properties via complex non-linear transformation, which makes it difficult to interpret. Computational imaging turns the qualitative intensity modulations into quantitative measurements of specimens' properties with inverse algorithms based on models of image formation. Quantitative phase imaging (*Popescu et al., 2006*; *Waller et al., 2010*; *Tian and Waller, 2015*) measures optical path length, that is, *specimen phase*, which reports density of the dry mass (*Barer, 1952*). Quantitative polarization microscopy in transmission mode reports angular anisotropy of the optical path length, that is, *retardance*, (*Inoue, 1953*; *Oldenbourg and Mei, 1995*; *Mehta et al., 2013*) and axis of anisotropy, that is, *orientation*, without label.

Quantitative label-free imaging measures intrinsic properties of the specimen and provides insights into biological processes that may not be obtained with fluorescence imaging. For example, Quantitative phase microscopy (*Park et al., 2018*) has been used to analyze membrane mechanics, density of organelles (*Imai et al., 2017*), cell migration, and recently fast propagation of action potential (*Ling et al., 2019*). Similarly, quantitative polarization microscopy has enabled discovery of the dynamic microtubule spindle (*Inoue, 1953*; *Keefe et al., 2003*), analysis of retrograde flow of F-actin network (*Oldenbourg et al., 2000*), imaging of white matter in adult human brain tissue slices (*Axer et al., 2011a*; *Axer et al., 2011b*; *Menzel et al., 2017*; *Mollink et al., 2017*; *Zeineh et al., 2017*; *Henssen et al., 2019*), and imaging of activity-dependent structural changes in brain tissue (*Koike-Tani et al., 2019*). Given the complementary information provided by specimen density and anisotropy, a joint imaging of phase and retardance has also been attempted (*Shribak et al., 2008*; *Ferrand et al., 2018*; *Baroni et al., 2020*). However, current methods for joint imaging of density and anisotropy are limited in throughput due to complexity of acquisition or can only be used for 2D imaging due to the lack of accurate 3D image formation models. We sought to develop a computational imaging method for joint measurements of phase and retardance of live 3D specimens with simpler light path and higher throughput.

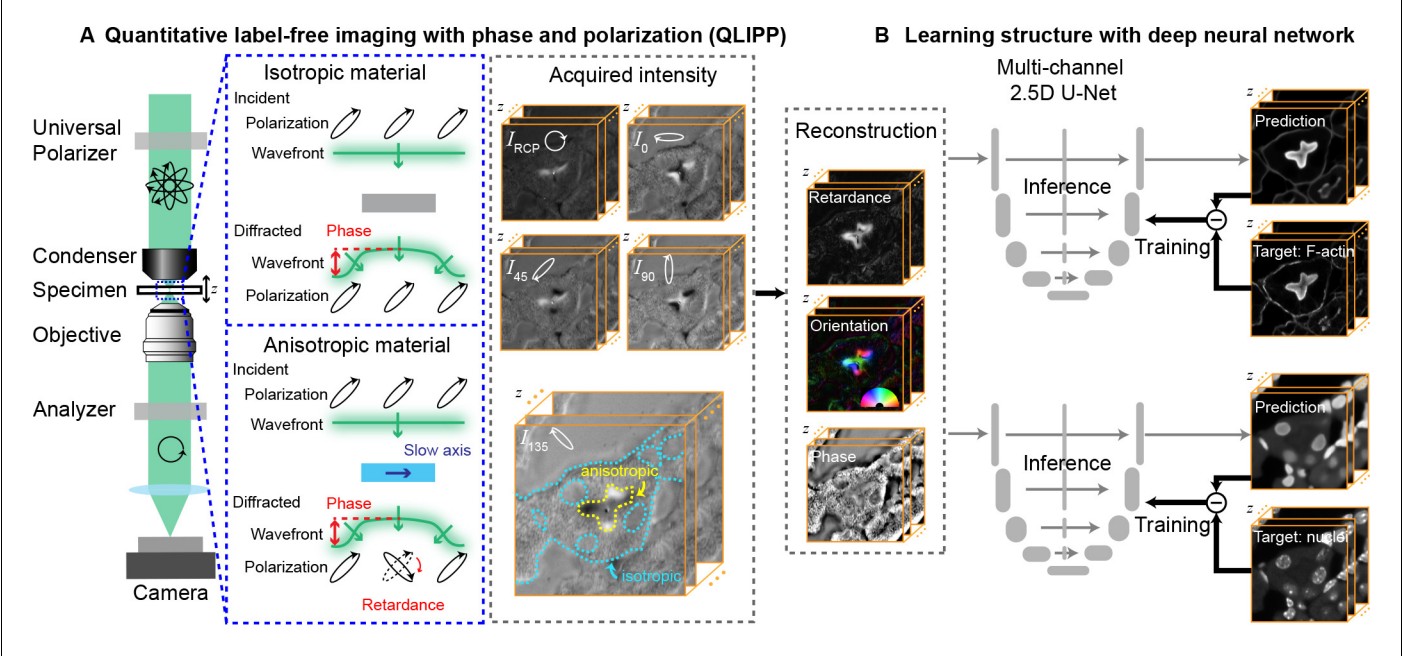

**Figure 1.** Measurements with QLIPP and analysis of structures with 2.5D U-Net. (**A**) Light path of the microscope. Volumes of polarization-resolved images are acquired by illuminating the specimen with light of diverse polarization states. Polarization states are controlled using a liquid-crystal universal polarizer. Isotropic material's optical path length variations cause changes in the wavefront (i.e., phase) of light that is measurable through defocused intensity stack. Anisotropic material not only changes the wavefront, but also changes the polarization of light depending on the degree of optical anisotropy (retardance) and orientation of anisotropy. Intensity Z-stacks of an example specimen, mouse kidney tissue, under five illumination polarization states ($I_{RCP}, I_0, I_{45}, I_{90}, I_{135}$ are shown. The intensity variations that encode the reconstructed physical properties of isotropic and anisotropic material are illustrated in the stack $I_{135}$. These polarization-resolved stacks are used to reconstruct (Materials and methods) the specimen's retardance, slow-axis orientation, and phase. Slow-axis orientation at given voxel reports the axis in the focal plane along which the material is the densest and is represent by a color according to the half-wheel shown in inset. (**B**) Multi-channel, 2.5D U-Net model is trained to predict fluorescent structures from label-free measurements. In this example 3D distribution of F-actin and nuclei are predicted. During training, pairs of label-free images and fluorescence images are supplied as inputs and targets, respectively, to the U-Net model. The model is optimized by minimizing the difference between the model prediction and the target. During inference, only label-free images are used as input to the trained model to predict fluorescence images.

In comparison to fluorescence measurements that provide molecular specificity, label-free measurements provide physical specificity. Obtaining biological insights from label-free images often requires identifying specific molecular structures. Recently, deep learning has enabled translation of qualitative and quantitative phase images into fluorescence images (*Ounkomol et al., 2018*; *Christiansen et al., 2018*; *Rivenson et al., 2018a*; *Rivenson et al., 2019*; *Lee et al., 2019*; *Petersen et al., 2017*). Among different neural network architecture, U-Net has been widely applied to image segmentation and translation tasks (*Ronneberger et al., 2015*; *Milletari et al., 2016*; *Ounkomol et al., 2018*; *Lee et al., 2019*). U-Net's success arises primarily from its ability to exploit image features at multiple spatial scales, and its use of skip connections between the encoding and decoding blocks. The skip connections give decoding blocks access to low-complexity, high-resolution features in the encoding blocks. In image translation, images from different modalities (label-free vs. fluorescence in our case) of the same specimen are presented to the neural network model. The neural network model learns the complex transformation from label-free to fluorescence images through the training process. The trained neural network model can predict fluorescence images from label-free images to enable analysis of distribution of a specific molecule. The accuracy with which the molecular structure can be predicted depends not just on the model, but also on the dynamic range and the consistency of the contrast with which the structure is seen in the label-free data. Some of the anisotropic structures are not visible in phase imaging data and therefore cannot be learned from phase imaging data. Reported methods of image translation have not utilized

optical anisotropy, which reports important structures such as cell membrane and axon bundles. Furthermore, previous work has mostly demonstrated prediction of single 2D fields of view. Volumetric prediction using 3D U-Net has been reported, but it is computationally expensive, such that downsampling the data at the expense of spatial resolution is required (*Ounkomol et al., 2018*). We sought to improve the accuracy of prediction of fluorescence images by using information contained in complementary measurements of density and anisotropy.

In this work, we report a combination of quantitative label-free imaging and deep learning models to identify biological structures from their density and anisotropy. First, we introduce quantitative label-free imaging with phase and polarization (QLIPP) that visualizes diverse structures by their phase, retardance, and orientation. QLIPP combines quantitative polarization microscopy (*Oldenbourg and Mei, 1995*; *Shribak and Oldenbourg, 2003*; *Mehta et al., 2013*) with the concept of phase from defocus (*Streibl, 1984*; *Waller et al., 2010*; *Streibl, 1985*; *Noda et al., 1990*; *Claus et al., 2015*; *Jenkins and Gaylord, 2015a*; *Jenkins and Gaylord, 2015b*; *Soto et al., 2017*), to establish a novel method for volumetric measurement of phase, retardance, and orientation (*Figure 1A*). Data generated with QLIPP can distinguish biological structures at multiple spatial and temporal scales, making it valuable for revealing the architecture of the postmortem archival tissue and organelle dynamics in live cells. QLIPP's optical path is simpler relative to earlier methods (*Shribak et al., 2008*), reconstruction algorithms are more accurate, and reconstruction software is open-source. QLIPP can be implemented on existing microscopes as a module and can be easily multiplexed with fluorescence. To translate 3D distribution of phase, retardance, and orientation to fluorescence intensities, we implement a computationally efficient multi-channel 2.5D U-Net architecture (*Figure 1B*) based on a previously reported single-channel 2.5D U-Net (*Han, 2017*). We use QLIPP for imaging axon tracts and myelination in archival brain tissue sections at two developmental stages. Label-free measurement of anisotropy allowed us to visualize axon orientations across whole sections. We demonstrate that QLIPP data increases accuracy of prediction of myelination in developing human brain as compared to brighfield data. Finally, we demonstrate robustness of the label-free measurements to experimental variations in labeling, which leads to more consistent prediction of myelination than possible with the experimental staining. Collectively, we propose a novel approach for imaging architectural order across multiple biological systems and analyzing it with a judicious combination of physics-driven and data-driven modeling approaches.

## Results

### QLIPP provides joint measurement of specimen density and anisotropy

The light path of QLIPP is shown in *Figure 1A*. It is a transmission polarization microscope based on computer controlled liquid crystal universal polarizer (*Oldenbourg and Mei, 1995*; *Shribak and Oldenbourg, 2003*; *Mehta et al., 2013*). QLIPP provides an accurate image formation model and corresponding inverse algorithm for simultaneous reconstruction of specimen phase, retardance, and slow axis orientation.

In QLIPP, specimens are illuminated with five elliptical polarization states for sensitive detection of specimens' retardance (*Shribak and Oldenbourg, 2003*; *Mehta et al., 2013*). For each illumination, we collect a Z-stack of intensity to capture specimens' phase information. Variations in the density of the specimen, for example lower density of nuclei relative to the cytoplasm, cause changes in refractive index and distort the wavefront of the incident light. The wavefront distortions lead to detectable intensity modulations through interference in 3D space as the light propagates along the optical axis. Intensity modulations caused by isopropic density variations (specimen phase) can be captured by acquiring a stack of intensities along the optical (Z) axis (*Streibl, 1984*; *Waller et al., 2010*). Anisotropic variations in the specimens' density result from alignment of molecules along a preferential axis, for example lipid membrane has higher anisotropy relative to the cytoplasm due to the alignment of lipid molecules. This anisotropic density variation (specimen retardance) induces polarization-dependent phase difference. Specimen retardance is often characterized by the axis along which anisotropic material is the densest (slow-axis) or by the axis perpendicular to it (fast-axis) (*de Campos Vidal et al., 1980*; *Salamon and Tollin, 2001*), and the difference in specimen phase between these two axes. In addition, multiple scattering by the specimen can reduce degree of polarization of light. The specimen retardance, slow-axis orientation, and degree of polarization

can be measured by probing the specimen with light in different polarization states. We develop a forward model of transformation using the formalism of partial polarization and phase transfer function to describe the relation between specimen physical properties and detected intensities. We then leverage above forward model to design an inverse algorithm that reconstructs quantitative specimen physical properties in 3D from the detected intensity modulations as illustrated in *Figure 1A*.

First, we utilize Stokes vector representation of partially polarized light (*Born and Wolf, 2013*; *Bass et al., 2009*; *Azzam, 2016*) to model the transformation from specimens' optical properties to acquired intensities (*Equation 7*). By inverting this transformation, we reconstruct 3D volumes of retardance, slow-axis orientation, brightfield, and degree of polarization. Proper background correction is crucial for detection of low retardance of the biological structures in the presence of high, non-uniform background resulting from the optics or imaging chamber. We use a two-step background correction method (Materials and methods) to correct the non-uniform background polarization (*Figure 2—figure supplement 2*). In addition to retardance and slow-axis orientation, our use of Stokes formalism enables reconstruction of brightfield and degree of polarization, in contrast to previous work that reconstructs just retardance and slow-axis orientation (*Shribak and Oldenbourg, 2003*; *Mehta et al., 2013*). The degree of polarization measures the fitness of our model with the experiment as explained later and the brightfield images enables reconstruction of specimen phase.

Second, we utilize phase transfer function formalism (*Streibl, 1985*; *Noda et al., 1990*; *Claus et al., 2015*; *Jenkins and Gaylord, 2015a*; *Jenkins and Gaylord, 2015b*; *Soto et al., 2017*) to model how 3D phase information is transformed into brightfield contrast (*Equation 17*). Specimen phase information is encoded in the brightfield images but in a complex fashion. In brightfield images, optically dense structures appear in brighter contrast than the background on one side of the focus, almost no contrast at the focus, and darker contrast than the background on the other side of the focus. This is illustrated by 3D brightfield images of nucleoli, the dense sub-nuclear domains inside nuclei (*Figure 2—video 1*). We invert our forward model to estimate specimen phase from 3D brightfield stack (*Equation 19*). Phase reconstruction from the brightfield volume shows nucleoli in positive contrast relative to background consistently as the nucleoli move through the focus (*Figure 2—video 1*). We note that the two-step background correction is essential for background-free retardance and orientation images, but not for phase image (*Figure 2—figure supplement 2*).

We illustrate wide applicability of QLIPP with images of human bone osteosarcoma epithelial (U2OS) cells, tissue section from adult mouse braintissue section from In the dividing U2OS cell (*Figure 2—video 2*, *Figure 2—video 3*), the phase image shows three-dimensional dynamics of dense cellular organelles, such as lipid vesicles, nucleoli, and chromosomes.The retardance and slow-axis orientation in U2OS cells (*Figure 2—video 2*, *Figure 2—video 3*) show dynamics of membrane boundaries, spindle, and lipid droplets. We note that the two-step background correction is essential to remove biases in the retardance and orientation images, but not for phase image (*Figure 2—figure supplement 2*). *Figure 2—video 3* shows that specific organelles can be discerned simply by color-coding the measured phase and retardance, illustrating that quantitative label-free imaging provides specificity to physical properties.

At larger spatial scale, the phase image identifies cell bodies and axon tracts in mouse and developing human brain tissue sections because of variations in their density. These density variations are more visible and interpretable in phase image as compared to the brightfield image (*Figure 2—figure supplement 3*). Axon tracts appear with noticeably high contrast in retardance and orientation images of mouse and human brain slices (*Figure 2*). The high retardance of the axons arises primarily from myelin sheath that has higher density perpendicular to the axon axis (*de Campos Vidal et al., 1980*; *Menzel et al., 2015*). Therefore, the slow axis of the axon tracts is perpendicular to the orientation of the tracts. . *Figure 2—figure supplement 4* and Figure 5 show stitched retardance and orientation images of a whole mouse brain slice, in which not only the white matter tracts, but also orientation of axons in cortical regions is visible. Note that the fine wavy structure in the right hemisphere of the slice is caused by sample preparation artifacts (*Figure 2—figure supplement 3*).

We show degree of polarization measurements in (*Figure 2—figure supplement 1*). The difference between retardance and degree of polarization is that retardance measures single scattering events within the specimen that alter the polarization of the light, but do not reduce the degree of

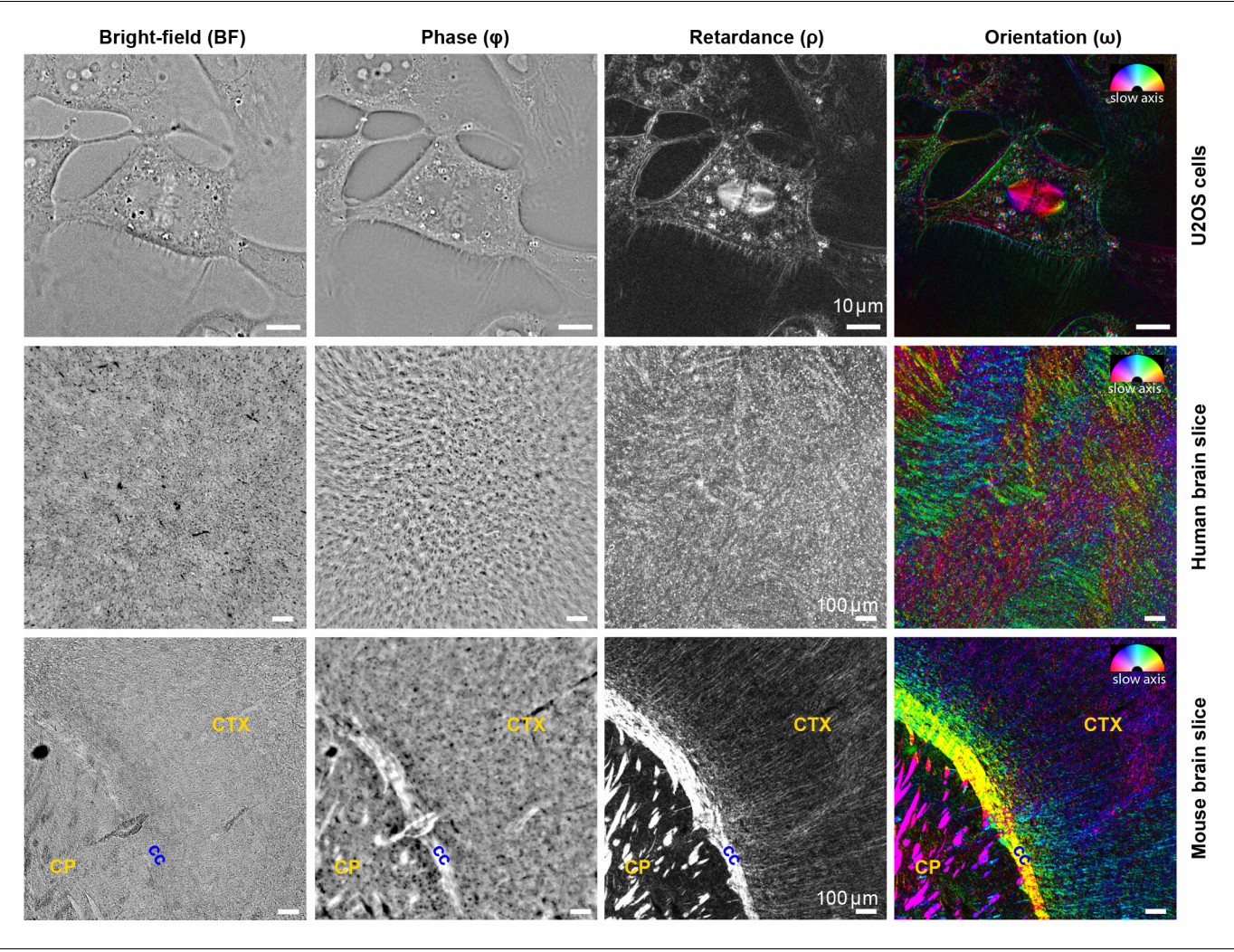

**Figure 2.** Complementary measurements of phase, retardance, and slow-axis orientation distinguish biological structures. Brightfield (*BF*), phase (Φ), retardance (ρ), and slow-axis orientation (ω) images of U2OS cells, human brain tissue, and adult mouse brain tissue are shown. In orientation images, slow axis and retardance of the specimen are represented by color (hue) and brightness, respectively. In U2OS cells, chromatin, lipid droplets, membranous organelles, and cell boundaries are visible in phase image due to variations in density, while microtubule spindle, lipid droplets, and cell boundaries are visible due to their anisotropy. In the adult mouse brain slices, axon tracts are more visible in phase, retardance, and orientation images compared to brightfield images, with slow axis perpendicular to the direction of the bundles (cc: corpus callosum, CP: caudoputamen, CTX: cortex). Similar label-free contrast variations are observed in developing human brain tissue slice, but with less ordered tracts compared to the adult mouse brain due to the early age of the brain. The 3D stack of live U2OS cell was acquired with 63 × 1.47 NA oil objective and 0.9 NA illumination, whereas images of mouse and human brain tissue were acquired with 10 × 0.3 NA air objective and 0.2 NA of illumination.

The online version of this article includes the following video and figure supplement(s) for figure 2:

**Figure supplement 1.** Degree of polarization (*DOP*) images for three specimens shown in *Figure 2*.

**Figure supplement 2.** Effect of background correction methods on reconstructed retardance and phase of the U2OS cell.

**Figure supplement 3.** Comparison of brightfield image (top) with quantitative phase image (bottom) of a mouse brain slice.

**Figure supplement 4.** Retardance (top) and orientation (bottom) measurements of a mouse brain slice, which report structural anisotropy and axon orientation (in physical line orientation), respectively.

**Figure 2—video 1.** Z-stacks of brightfield and phase images of U2OS cells.

https://elifesciences.org/articles/55502#fig2video1

**Figure 2—video 2.** Time-lapse of phase, retardance, and slow axis orientation in a dividing U2OS cell shows differences in density and anisotropy of organelles.

https://elifesciences.org/articles/55502#fig2video2

**Figure 2—video 3.** 3D rendering of the time-lapse showing diverse structures color coded by their retardance and phase in U2OS cells shown in *Figure 2*.

https://elifesciences.org/articles/55502#fig2video3

polarization. On the other hand, low degree of polarization indicates multiple scattering events that reduce the polarization of light and thus mismatch of the specimen optical properties from the model assumptions. In the future, we plan to pursue models that account for diffraction and scattering effects in polarized light microscopy that would enable more precise retrieval of specimen properties.

Data reported above illustrate simultaneous and quantitative measurements of density, structural anisotropy, and orientation in 3D biological specimens, for the first time to our knowledge. The Python software for QLIPP reconstruction is available at https://github.com/mehta-lab/reconstruct-order. In the next sections, we discuss how these complementary label-free measurements enable prediction of fluorescence images and analysis of architecture.

## 2.5D U-Net allows efficient prediction of fluorescent structures from multi-channel label-free images

In contrast to fluorescence imaging, label-free measurement of density and anisotropy visualize several structures simultaneously but individual structures can be difficult to identify. Label-free measurements are affected by the expression of specific molecules, but do not report the expression directly. To obtain images of specific molecular structures from QLIPP data, we optimized convolutional neural network models to translate 3D label-free stacks into 3D fluorescence stacks.

Proper prediction of of fluorescent structures with deep learning requires joint optimization of image content, architecture of the neural network, and the training process. The optimization led us to a residual 2.5D U-Net that translates a small stack (5–7 slices) of label-free channels to the central slice of fluorescent channel throughout 3D volume. We use images of the mouse kidney tissue section as a test dataset for optimizing the model architecture and training strategies. We chose the mouse kidney tissue section because it has both anisotropic and isotropic structures (F-actin and nuclei). Additionally, both structures are robustly labeled with no noticeable artifacts. Later we demonstrate predicting the fluorescent labels in specimen where labeling is not robust (Figure 6).

## Optimization of 2.5D model for prediction of fluorescence images

Our work builds upon earlier work (*Ounkomol et al., 2018*) on predicting fluorescence stacks from brightfield stacks using 3D U-Net. *Ounkomol et al., 2018* showed fluorescence predicted by 3D U-Net is superior than 2D U-Net. However, applying 3D U-Net to microscopy images poses a few limitations. Typical microscopy stacks are bigger in their extent in the focal plane (~2000 × 2000 pixels) and smaller in extent along the optical axis (usually <40 Z slices). Since the input is isotropically downsampled in the encoding path of the 3D U-Net, it requires sufficiently large number of Z slices to propagate the data through encoding and decoding blocks. As an example, for a minimum of 3 layers in U-Net and 16 pixels at the end of the encoder path, one will need at least 64 Z slices

**Table 1.** Accuracy of 3D prediction of F-actin from retardance stack using different neural networks.

Above table lists median values of the Pearson correlation (*r*) and structural similarity index (SSIM) between prediction and target F-actin volumes. We report accuracy metrics for Slice→Slice (2D) ,Stack→Slice (2.5D), and Stack→Stack (3D) models trained to predict F-actin from retardance using Mean Absolute Error (MAE or L1) loss. We segmented target images with a Rosin threshold to discard tiles that mostly contained background pixels. To dissect the differences in prediction accuracy along and perpendicular to the focal plane, we computed (Materials and methods) test metrics separately over XY slices ($r_{xy}$, $SSIM_{xy}$) and XZ slices ($r_{xy}$, $SSIM_{xz}$) of the test volumes, as well as over entire test volumes ($r_{xyz}$, $SSIM_{xyz}$). Best performing model according to each metric is displayed in bold.

| Translation model | Input(s) | $r_{xy}$ | $r_{xz}$ | $r_{xyz}$ | $SSIM_{xy}$ | $SSIM_{xz}$ | $SSIM_{xyz}$ |
|---|---|---|---|---|---|---|---|
| Slice→Slice (2D) | ρ | 0.82 | 0.79 | 0.83 | 0.78 | 0.71 | 0.78 |
| Stack→Slice (2.5D, $z = 3$) | ρ | 0.85 | 0.83 | 0.86 | 0.80 | 0.75 | 0.81 |
| Stack→Slice (2.5D, $z = 5$) | ρ | 0.86 | 0.84 | **0.87** | 0.81 | 0.76 | 0.82 |
| Stack→Slice (2.5D, $z = 7$) | ρ | **0.87** | **0.85** | **0.87** | **0.82** | **0.77** | 0.83 |
| Stack→Stack (3D, $z = 96$) | ρ | 0.86 | 0.84 | 0.86 | **0.82** | 0.76 | **0.85** |

**Table 2.** Accuracy of prediction of F-actin in mouse kidney tissue as a function of input channels.
Median values of the Pearson correlation ($r$) and structural similarity index (SSIM) between predicted and target volumes of F-actin. We evaluated combinations of brightfield (BF), phase (Φ), retardance (ρ), orientation x ($\omega_x$), and orientation y ($\omega_y$), as input. Model training conditions and computation of test metrics is described in *Table 1*.

| Translation model | Input(s) | $r_{xy}$ | $r_{xz}$ | $r_{xyz}$ | $SSIM_{xy}$ | $SSIM_{xz}$ | $SSIM_{xyz}$ |
|---|---|---|---|---|---|---|---|
| Stack→Slice (2.5D, $z = 5$) | ρ | 0.86 | 0.84 | 0.87 | 0.81 | 0.76 | 0.82 |
| | BF | 0.86 | 0.84 | 0.86 | 0.82 | 0.77 | 0.83 |
| | Φ | 0.87 | 0.85 | 0.88 | **0.83** | 0.78 | 0.84 |
| | Φ, ρ, $\omega_x$, $\omega_y$ | **0.88** | **0.87** | **0.89** | **0.83** | **0.80** | **0.85** |
| | BF, ρ, $\omega_x$, $\omega_y$ | **0.88** | **0.87** | **0.89** | **0.83** | 0.79 | **0.85** |

(*Figure 3—figure supplement 1*). Therefore, the use of 3D translation models often requires upsampling of the data in Z, which increases data size and makes training 3D translation model computationally expensive.

To reduce the computational cost without losing accuracy of prediction, we evaluated the prediction accuracy as a function of model dimensions for a highly ordered, anisotropic structure (F-actin) and for less ordered, isotropic structure (nuclei) in mouse kidney tissue. In mouse kidney tissue, the retardance image highlights capillaries within glomeruli, and brush borders in convoluted tubules, among other components of the tissue. The nuclei appear in darker contrast in the retardance image, because of the isotropic architecture of chromatin. We evaluated three model architectures to predict fluorescence volumes: slice→slice (2D in short) models that predict 2D fluorescence slices from corresponding 2D label-free slices, stack→slice (2.5D in short) models that predict the central 2D fluorescence slice from a stack of adjacent label-free slices, and stack→stack (3D in short) models that predict 3D fluorescent stacks from label-free stacks. For 2.5D models, 3D translation is achieved by predicting one 2D fluorescence plane per stack (z = 3, 5, 7) of label-free inputs. We added a residual connection between the input and output of each block to speed up model training (*Milletari et al., 2016*; *Drozdzal et al., 2016*).

In order to fit 3D models on the GPU, we needed to predict overlapping sub-stacks, which were stitched together to get the whole 3D stack ( see Materials and methods and *Figure 3—figure supplement 1* for the description of the network architecture and training process). We used Pearson correlation coefficient and structural similarity index (SSIM) (*Wang and Bovik, 2009*) between predicted fluorescent stacks and target fluorescent stacks to evaluate the performance of the models (Materials and methods). We report these metrics on the test set (*Table 1*, *Table 2*, *Table 3*), which was not used during the training.

The predictions with 2D models show discontinuity artifacts along the depth (*Figure 3*, *Figure 3— video 2*), as also observed in prior work (*Ounkomol et al., 2018*). The 3D model predicts smoother structures along the Z dimension with improved prediction in the XY plane. 2.5D model shows prediction accuracy comparable to 3D model, with higher prediction accuracy as the number of z-slices

**Table 3.** Accuracy of prediction of nuclei in mouse kidney tissue.
Median values of the Pearson correlation ($r$) and structural similarity index (SSIM) between predicted and target volumes of nuclei. See *Table 2* for description.

| Translation model | Input(s) | $r_{xy}$ | $r_{xz}$ | $r_{xyz}$ | $SSIM_{xy}$ | $SSIM_{xz}$ | $SSIM_{xyz}$ |
|---|---|---|---|---|---|---|---|
| Stack→Slice (2.5D, $z = 5$) | ρ | 0.84 | 0.85 | 0.85 | 0.81 | 0.76 | 0.82 |
| | BF | 0.87 | 0.88 | 0.87 | 0.82 | 0.77 | 0.84 |
| | Φ | 0.88 | 0.88 | 0.88 | 0.83 | 0.78 | 0.85 |
| | Φ, ρ, $\omega_x$, $\omega_y$ | **0.89** | 0.89 | **0.89** | **0.84** | **0.80** | **0.86** |
| | BF, ρ, $\omega_x$, $\omega_y$ | **0.89** | **0.90** | **0.89** | **0.84** | **0.80** | **0.86** |

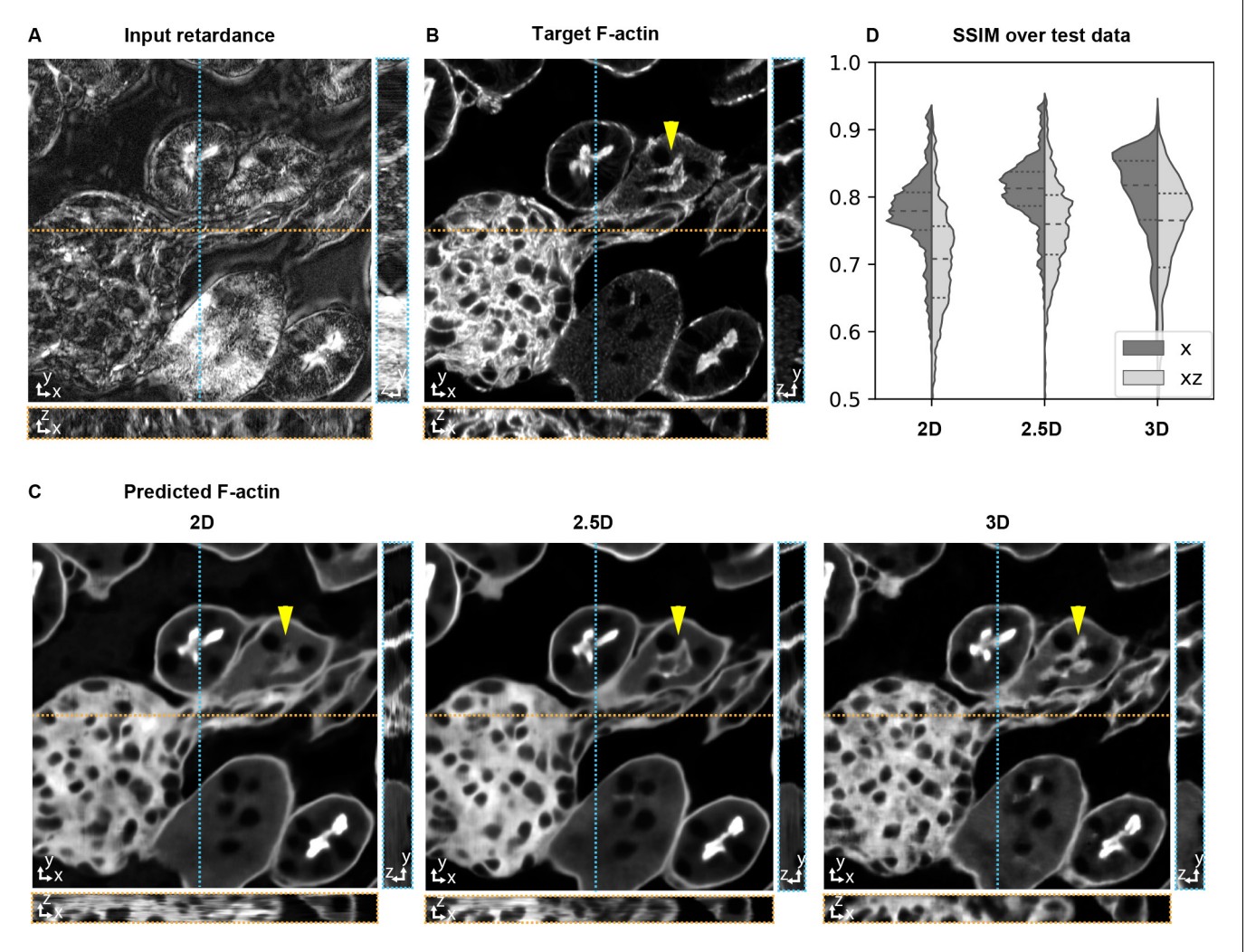

**Figure 3.** Accuracy of 3D prediction with 2D, 2.5D, and 3D U-Nets. Orthogonal sections (XY - top, XZ - bottom, YZ - right) of a glomerulus and its surrounding tissue from the test set are shown depicting (**A**) retardance (input image), (**B**) experimental fluorescence of F-actin stain (target image), and (**C**) Predictions of F-actin (output images) using the retardance image as input with different U-Net architectures. (**D**) Violin plots of structral-similarty metric (SSIM) between images of predicted and target stain in XY and XZ planes. The horizontal dashed lines in the violin plots indicate 25th quartile, median, and 75th quartile of SSIM. The yellow triangle in C highlights a tubule structure, whose prediction can be seen to improve as the model has access to more information along Z. The same field of view is shown in *Figure 3—video 1*, *Figure 3—video 2*, and *Figure 4—video 1*.

The online version of this article includes the following video and figure supplement(s) for figure 3:

**Figure supplement 1.** Schematic illustrating U-Net architectures.

**Figure 3—video 1.** Z-stacks of brightfield, phase, retardance, and orientation images of mouse kidney tissue.

https://elifesciences.org/articles/55502#fig3video1

**Figure 3—video 2.** Through focus series showing 3D F-actin distribution in the test field of view shown in *Figure 3*.

https://elifesciences.org/articles/55502#fig3video2

in the 2.5D model input increases. (*Figure 3C and D*; *Table 1*; *Figure 3—video 2*). While 2.5D model shows similar performance to 3D model, we note that we could train the 2.5D model with ~3× more parameters than 3D model (Materials and methods) in shorter time. In our experiments, training a 3D model with 1.5*M* parameters required 3.2 days, training a 2D model with 2*M* parameters required 6 hr, and training a 2.5D model with 4.8*M* parameters and five input z-slices required 2 days, using ~100 training volumes. This is because the large memory usage of 3D model significantly limits its training batch size and thus the training speed.

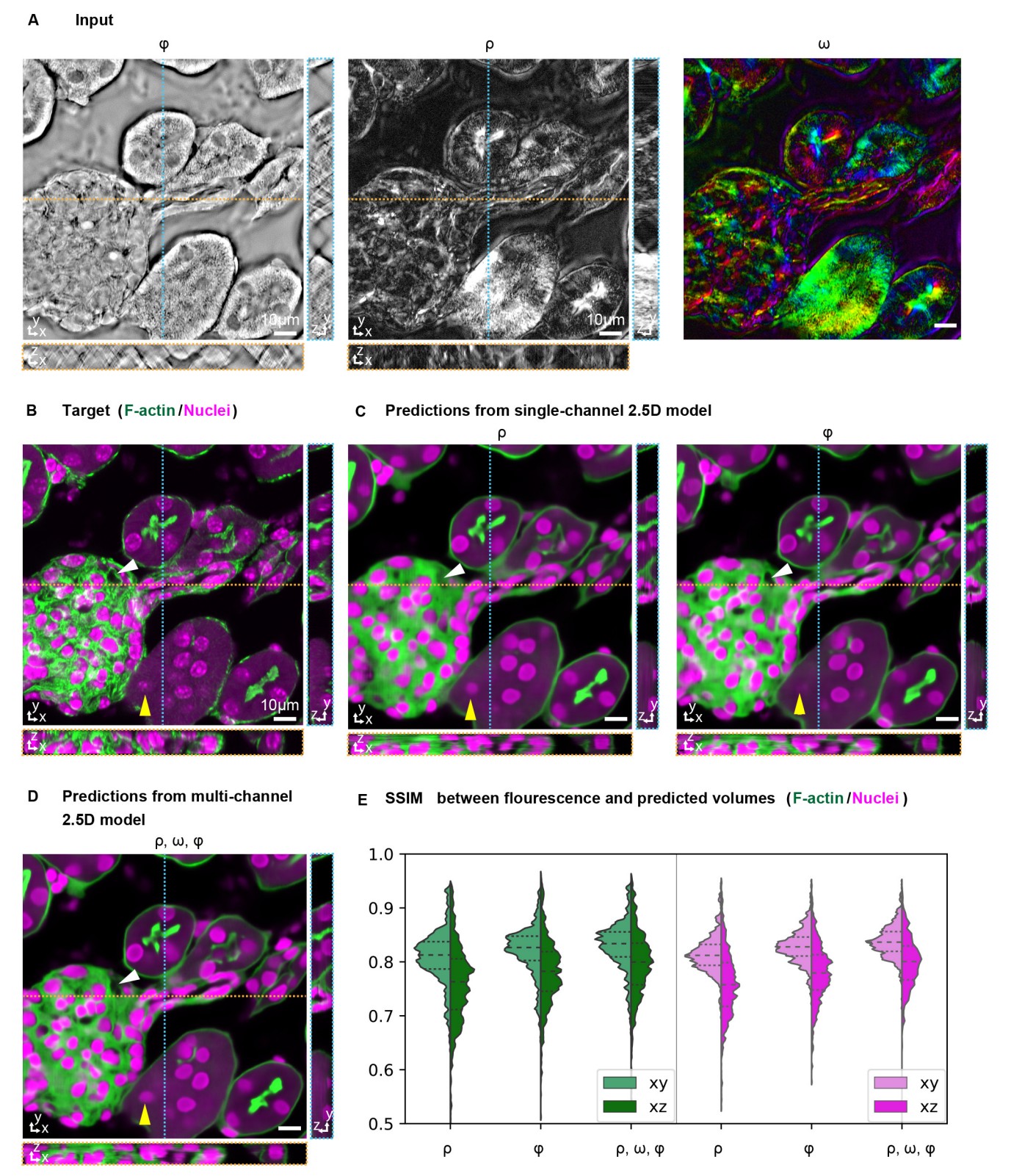

**Figure 4.** Prediction accuracy improves with multiple label-free contrasts as inputs. 3D predictions of ordered F-actin and nuclei from different combinations of label-free contrasts using the 2.5D U-Net model. (A) Label-free measurements used as inputs for model training: retardance (ρ), phase (Φ), and slow axis orientation (ω). (B) The corresponding 3D volume showing the target fluorescent stains. Phalloidin-labeled F-actin in shown green and
*Figure 4 continued on next page*

*Figure 4 continued*

DAPI labeled nuclei is shown in magenta. (C) F-actin and nuclei predicted with single channel models trained on retardance (ρ) and phase (Φ) alone are shown. (D) F-actin and nuclei predicted with multi-channel models trained with the combined input of retardance, orientation, and phase. The yellow triangle and white triangle point out structures missing in predicted F-actin and nuclei distributions when only one channel is used as an input, but predicted when all channels are used. (E) Violin plots of structral-similarty metric (SSIM) between images of predicted and experimental stain in XY and XZ planes. The horizontal dashed lines indicate 25th quartile, median, and 75th quartile of SSIM. The 3D label-free inputs used for prediction are shown in *Figure 3—video 1*.

The online version of this article includes the following video and figure supplement(s) for figure 4:

**Figure supplement 1.** Pearson correlation and SSIM are insensitive to small structural differences in the images.
**Figure supplement 2.** Fine structural features are better predicted with foreground loss.
**Figure 4—video 1.** Through focus series showing 3D F-actin and nuclei distribution in the test field of view shown in *Figure 4*.
https://elifesciences.org/articles/55502#fig4video1

The Python code for training our variants of image translation models is available at https://github.com/czbiohub/microDL.

## Predicting structures from multiple label-free contrasts improves accuracy

Considering the trade-off between computation speed and model performance, we adopt 2.5D models with five input Z-slices to explore how combinations of label-free inputs affect the accuracy of prediction of fluorescent structures.

We found that when multiple label-free measurements are jointly used as inputs, both F-actin and nuclei are predicted with higher fidelity compared to when only a single label-free measurement is used as the input (*Table 2* and *Table 3*). *Figure 4C–D* shows representative structural differences in the predictions of the same glomerulus as *Figure 3*. The continuity of prediction along Z-axis improves as more label-free contrasts are used for prediction (*Figure 4—video 1*). These results indicate that our model leverages information in complementary physical properties to predict target structures. We note that using complementary label-free contrasts boosts the performance of 2.5D models to exceed the performance of 3D single-channel models without significantly increasing the computation cost (compare *Table 1* and *Table 2*). Noticeably, fine F-actin bundles have been shown challenging to predict from single label-free input. We found fine F-actin bundles can be predicted from multiple label-free inputs when the model is trained to minimize the difference between the fluorescence target and prediction over only the foreground pixels in the image (*Figure 4—figure supplement 2*).

Interestingly, when only a single contrast is provided as the input, a model trained on phase images has higher prediction accuracy than the model trained on brightfield images. This is possibly because the phase image has consistent, quantitative contrast along z-axis, while the depth-dependent contrast in brightfield images makes the translation task more challenging. This improvement of using phase over brightfield images, however, is not observed when the retardance and orientation are also included as inputs. This is possibly because quantitative retardance and orientation complement the qualitative brightfield input and simplify the translation task.

In conclusion, above results show that 2.5D multi-contrast models predict 3D structures with superior accuracy than single channel 3D U-Net models, but have multiple practical advantages that facilitate scaling of the approach. In addition, the results show that structures of varying density and order can be learned with higher accuracy when complementary physical properties are combined as inputs.

## Imaging architecture of mouse and human brain tissue with QLIPP

Among electron microscopy, light microscopy, and magnetic resonance based imaging of brain architecture, the resolution and throughput of light-microscopy provides the ability to image whole brain slices at single axon resolution in a reasonable time (*Kleinfeld et al., 2011*; *Axer et al., 2011a*; *Axer et al., 2011b*; *Menzel et al., 2017*; *Mollink et al., 2017*; *Zeineh et al., 2017*; *Henssen et al., 2019*). Light-microscopy is also suitable for imaging biological processes while brain tissue is kept

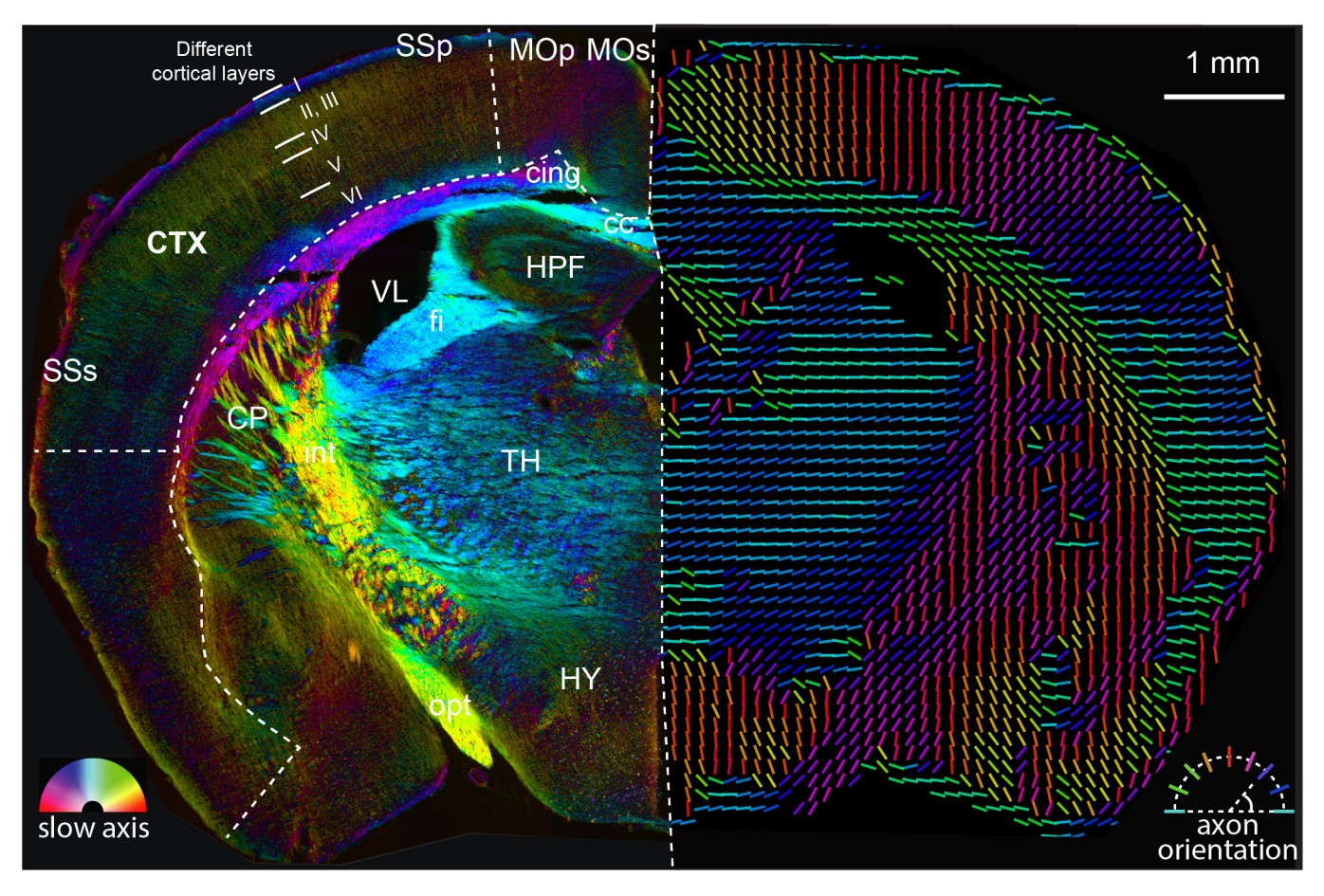

**Figure 5.** Analysis of anatomy and axon orientation of an adult mouse brain tissue with QLIPP. The retardance and orientation measurements are rendered with two approaches in opposing hemispheres of the mouse brain, respectively. In the left panel, the slow-axis orientation is displayed with color (hue) and the retardance is displayed with brightness as shown by the color legend in bottom-left. In the right panel, the colored lines represent fast axis and the direction of the axon bundles in the brain. The color of the line still represents the slow-axis orientation as shown by color legend in bottom-right. Different cortical layers and anatomical structures are visible through this measurements. This mouse brain section is a coronal section at around bregma −1.355 mm and is labeled according to Allen brain reference atlas (level 68) (*Lein et al., 2007*). cc: corpus callosum, cing: cingulum bundle, CTX: cortex, CP: caudoputamen, fi: fimbria, HPF: hippocampal formation, HY: hypothalamus, int: internal capsule, MOp: primary motor cortex, MOs: secondary motor cortex, opt: optic tract, SSp: primary somatosensory area, SSs: supplemental somatosensory area, TH: thalamus, VL: lateral ventricle.

The online version of this article includes the following figure supplement(s) for figure 5:

**Figure supplement 1.** The full-size mouse brain images of two rendering approaches shown in *Figure 5*.

alive (*Ohki et al., 2005*; *Koike-Tani et al., 2019*). With quantitative imaging of brain architecture and activity at light resolution, one can envision the possibility of building probabilistic models that relate connectivity and function. QLIPP's high-resolution, quantitative nature, sensitivity to low anisotropy of gray matter (*Figure 2*), and throughput make it attractive for imaging the architecture and activity in brain slices. Here, we explore how QLIPP can be used to visualize the architecture of the sections of adult mouse brain and archival sections of prenatal human brain.

## Adult mouse brain tissue

We first imaged an adult mouse brain tissue section located at bregma −1.355 mm (level 68 in Allen brain reference atlas [*Lein et al., 2007*]) with QLIPP and rendered retardance and slow-axis orientation in two ways as shown in *Figure 5*. The left panel renders the measured retardance in brightness

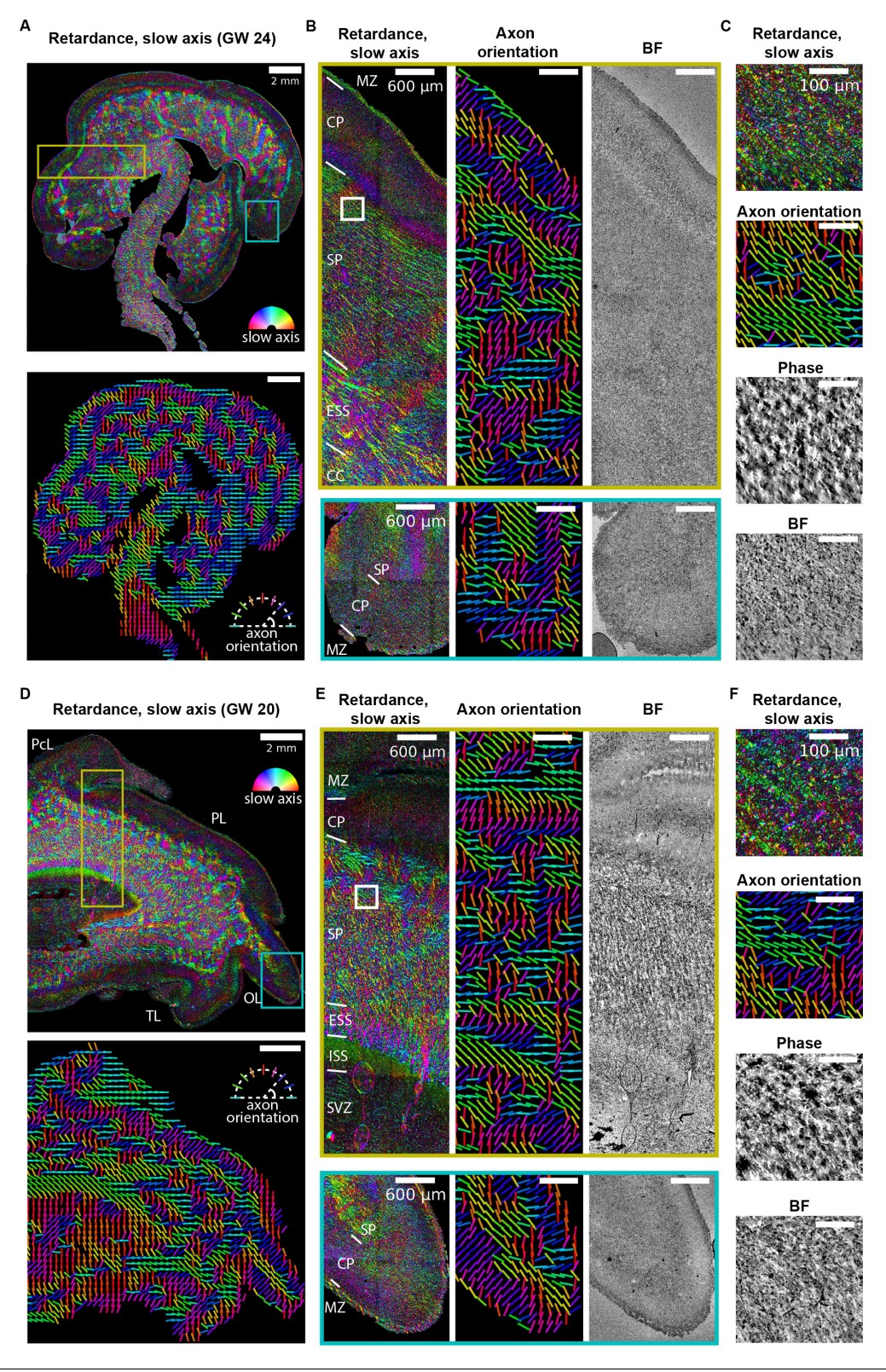

**Figure 6.** Label-free mapping of axon tracts in developing human brain tissue section. (A) (top) Stitched image of retardance and slow axis orientation of a gestational week 24 (GW24) brain section from the test set. The slow axis orientation is encoded by color as shown by the legend. (Bottom) Axon orientation indicated by the lines. (B) Zoom-ins of retardace + slow axis, axon orientation, and brightfield at brain regions indicated by the yellow and cyan boxes in (A). (C) Zoom-ins of label-free images at brain regions indicated by the white box in (B) (D–F) Same as (A–C), but for GW20 sample. MZ:

*Figure 6 continued on next page*

*Figure 6 continued*

marginal zone; CP: cortical plate; SP: subplate; ESS: external sagittal stratum; ISS: internal sagittal stratum; CC: corpus callosum; SVZ: subventricular zone; PcL: paracentral lobule PL: parietal lobe; OL: occipital lobe; TL: temporal lobe. Anatomical regions in (**B**, **D**, and **E**) are identified by referencing to developing human brain atlas (*Bayer and Altman, 2003*).

The online version of this article includes the following figure supplement(s) for figure 6:

**Figure supplement 1.** Brightfield and phase images of human brain sections.

and slow-axis orientation in color, highlighting anatomical features of all sizes. The right panel renders the fast-axis orientation of the mouse brain section (orthogonal to the slow-axis orientation) as colored lines. It has been shown (*de Campos Vidal et al., 1980*; *Menzel et al., 2015*) that when axons are myelinated, the slow axis is perpendicular to the axon axis, while the fast axis is parallel to it. The visualization in the right panel highlights meso-scale axon orientation in the mouse brain tissue with spatial resolution of ∼ 100 μm, that is, each line represents net orientation of the tissue over the area of ∼ 100 μm × 100 μm. The full section rendered with both approaches is shown in *Figure 5—figure supplement 1*.

By comparing the size and optical measurements in our label-free images against Allen brain reference atlas, we are able to recognize many anatomical landmarks. For example, the corpus callosum (cc) traversing the left and right hemispheres of the brain is a highly anisotropic bundle of axons. The cortex (CTX) is the outermost region of the brain, with axons projecting down towards the corpus callosum and other sub-cortical structures. Within the inner periphery of the corpus callosum, we can identify several more structures such as hippocampus (HPF), lateral ventricle (VL), and caudoputamen (CP). With these evident anatomical landmarks, we are able to reference to Allen brain reference atlas (*Lein et al., 2007*) and label more anatomical areas of the brain such as the sensory (SSp, SS) and motor (MOp, MOs) cortical areas.

We also found that six cortical layers are distinguishable in terms of strength of the retardance signal and the orientational pattern. These data are consistent with reports that layer I contains axon bundles parallel to the cortical layer (*Zilles et al., 2016*). Layer VI contains axon bundles that feed to and from the corpus callosum, so the orientation of the axon is not as orthogonal to the cortical layers as the axons in the other layers. The retardance signal arises from the collective anisotropy of myelin sheath wrapping around axons. Layers IV and V contain higher density of cell bodies and correspondingly lower density of the axons, leading to lower signal in retardance.

## Tissue from developing human brain

We next imaged brain sections from developing human samples of two different ages, gestational week 24 (GW 24) (*Figure 6A–C*, *Figure 6—figure supplement 1A*) and GW20 (*Figure 6D–F*, *Figure 6—figure supplement 1A*) which correspond to the earliest stages of oligodendrocyte maturation and early myelination in the cerebral cortex (*Jakovcevski et al., 2009*; *Miller et al., 2012*; *Snaidero and Simons, 2014*). Similar to the observations in the mouse brain section (*Figure 5*, *Figure 2—figure supplement 4*), the stitched retardance and orientation images show both morphology and orientation of the axon tracts that are not accessible with brightfield or phase imaging, with fast axis orientation parallel to the axon axis. The retardance in subplate is higher than cortical plates at both time points, which is consistent with the reduced myelin density in the cortical plate relative to the white matter. Importantly, with our calibration and background correction procedures (Materials and methods), our imaging approach has the sensitivity to detect axon orientation in the developing cortical plate, despite the lower retardance in developing brain compared to adult brain due to the low myelination in early brain development (*Miller et al., 2012*; *Snaidero and Simons, 2014*). Different cortical layers are visible in the retardance and orientation images at both time points. With this approach, we could identify different anatomical structures in the developing human brain without additional stains by referencing to the developing human brain atlas (*Bayer and Altman, 2003*, *Figure 6*). The individual axon tracts are also visible in phase image while with lower contrast as the phase image measures the density variation but not the axon orientation.

To analyze the variations in the density of the human brain tissue, we reconstructed 2D phase, unlike 3D phase reconstruction for U2OS cells (*Figure 2*) and kidney tissue (*Figure 4*). The archival tissue was thinner (12 μm thick) than the depth of field (∼16 μm) of the low magnification objective

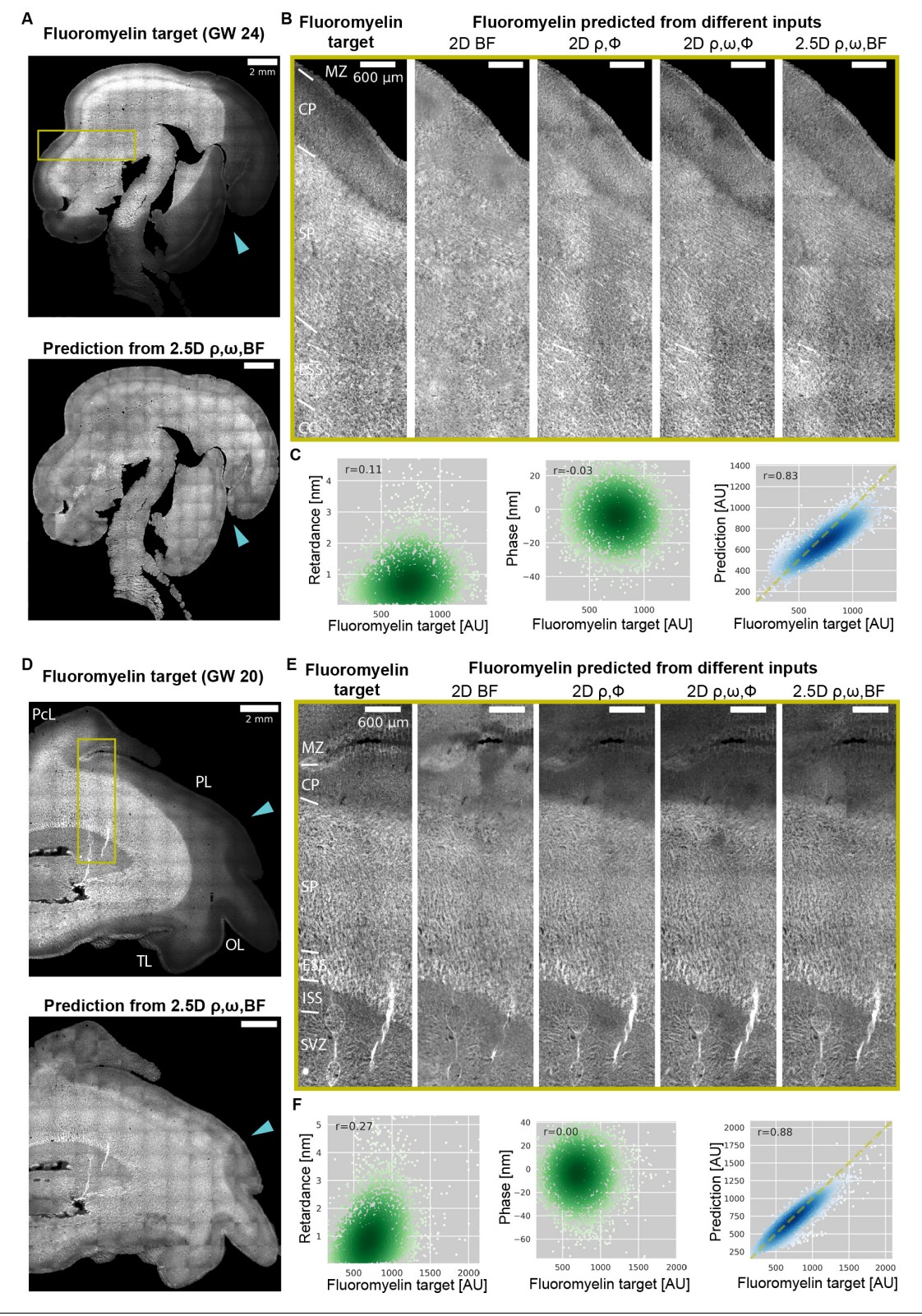

**Figure 7.** Prediction of myelination in developing human brain from QLIPP data and rescue of inconsistent labeling. (**A**) Stitched image of experimental FluoroMyelin stain of the same (GW24) brain section from the test set (top) and FluoroMyelin stain predicted from retardance, slow axis orientation, brightfield by the 2.5D model (bottom). The cyan arrow head indicates large staining artifacts in the experimental FluoroMyelin stain but rescued in model prediction. (**B**) Zoom-ins of experimental and predicted FluoroMyelin stain using different models at brain regions indicated by the

*Figure 7 continued on next page*

*Figure 7 continued*

yellow box in (**A**) rotated by 90 degrees. From left to right: experimental FluoroMyelin stain; prediction from brightfield using 2D model; prediction from retardance and phase using 2D model; prediction from retardance, phase, and orientation using 2D model; prediction from retardance, brightfield, and orientation using 2.5D model. (**C**) From region shown in (**B**) we show scatter plot and Pearson correlation of target FluoroMyelin intensity v.s. retardace (left), phase (middle), FluoroMyelin intensity predicted from retardance, brightfield, and orientation using 2.5D model (right). Yellow dashed line indicates the function y = x. (**D–F**) Same as (**A–C**), but for GW20 sample. MZ: marginal zone; CP: cortical plate; SP: subplate; ESS: external sagittal stratum; ISS: internal sagittal stratum; CC: corpus callosum; SVZ: subventricular zone; PcL: paracentral lobule PL: parietal lobe; OL: occipital lobe; TL: temporal lobe.

The online version of this article includes the following figure supplement(s) for figure 7:

**Figure supplement 1.** Normalizing training data per dataset yields prediction with correct dynamic range of intensity.

**Figure supplement 2.** Model predicted FluoroMyelin intensity becomes more accurate as more label-free channels are included as input.

(10X) we used for imaging large areas. *Figure 6B,C,E and F* show the retardance, slow-axis orientation, axon orientation, brightfield, and phase images. Major regions such as the subplate and cortical plate can be identified in both samples. While density information represented by brightfield and phase images can identify some of the anatomical structures, axon-specific structures can be better identified with measurements of anisotropy.

To our knowledge, the above data are the first report of label-free imaging of architecture and axon tract orientation in prenatal brain tissue. The ability to resolve axon orientation in the cortical plate of the developing brain, which exhibits very low retardance, demonstrates the sensitivity and resolution of our approach.

## Predicting myelination in sections of developing human brain

Next, we explore how information in the phase and retardance measurements can be used to predict myelination in prenatal human brain. The human brain undergoes rapid myelination during late development as measured with magnetic resonance imaging (MRI) (*Heath et al., 2018*). Interpretation of the myelination from MRI contrast requires establishing its correlation with histological measurements of myelin levels (*Khodanovich et al., 2019*). Robust measurements of myelination in postmortem human brains can provide new insights in myelination of human brain during development and during degeneration. QLIPP data in *Figure 6* indicate that label-free measurements are predictive of the level of myelination but relationship among them is complex (*Figure 7C and F*). We employed our multi-channel 2D and 2.5D U-Net models to learn the complex transformation from label-free contrasts to myelination. Importantly, we developed a data normalization and training strategy that enables prediction of myelination across large slices and multiple developmental time points. We also found that a properly trained model can rescue inconsistencies in fluorescent labeling of myelin, which is often used as histological groundtruth.

**Table 4.** Accuracy of prediction of FluoroMyelin in human brain tissue slices across two developmental points (GW20 and GW24). Median values of the Pearson correlation (*r*) and structural similarity index (SSIM) between predictions of image translation models and target fluorescence. We evaluated combinations of retardance ($\rho$), orientation x ($\omega_x$), orientation y ($\omega_y$), phase ($\Phi$), and brightfield (*BF*) as inputs. These metrics are computed over 15% of the fields of view from two GW20 datasets and two GW24 test datasets. The 2D models take ~ 4 hours to converge, whereas 2.5D models take ~ 64 hours to converge.

| Translation model | Input(s) | $r_{xy}$ | $SSIM_{xy}$ |
|---|---|---|---|
| Slice→Slice (2D) | BF | 0.72 | 0.71 |
| | $\rho$, $\Phi$ | 0.82 | 0.82 |
| | $\rho$, $\omega_x$, $\omega_y$, $\Phi$ | 0.86 | **0.85** |
| Stack→Slice (2.5D, $z = 5$) | BF, $\rho$, $\omega_x$, $\omega_y$ | **0.87** | **0.85** |

## Data pooling for prediction over large sections of prenatal human brain

In order to train the model, we measured the level of myelination with FluoroMyelin, a lipophilic dye that can stain myelin without permeabilization (*Monsma and Brown, 2012*). We found the detergents used in most permeabilization protocols remove myelin from the tissue and affect our label-free measurements. We trained multi-contrast 2D and 2.5D models with different combinations of label-free input contrasts and FluoroMyelin as the target to predict. To avoid overfitting and build a model that generalizes to different developmental ages and different types of sections of the brain, we pooled imaging datasets from GW20 and GW24 with two different brain sections for each age. The pooled dataset was then split into training, validation, and test set. Similar to the observations in the mouse kidney tissue, the prediction accuracy improves as more label-free contrasts are included in training but with higher accuracy gain compared to the mouse kidney tissue. This is most likely because the additional information provided by adding more label-free channels is more informative for the model to predict the more complex and variable of human brain structures. On the other hand, 2.5D model with all four input channels shows similar performance as 2D model for this dataset due to the relatively large depth of field (~16 μm) compared to the sample thickness (12 μm thick), so additional Z-slices only provide phase information but no extra structural information along the z dimension. (*Table 4*).

To test the accuracy of prediction over large human brain slices that span multiple fields of view, we predicted FluoroMyelin using label-free images of whole sections from GW24 and GW20 brains that were not used for model training or validation. We ran model inference on each field of view and then stitched the predicted images together to obtain a stitched prediction with 20,000 × 20,000 pixels (*Figure 7A and D*). To the best of our knowledge, these are the largest predicted fluorescence image of tissue sections that have been generated. We were able to predict myelination level in sections from both time points with a single model, with increasing accuracy as we included more label-free channels as the input, similar to our observations from the test dataset of the mouse kidney slice (*Table 4* and *Figure 7B and E*). The scatter plots of pixel intensities show that model-predicted FluoroMyelin intensities correlate with the target FluoroMyelin stain significantly better than the label-free contrasts alone (*Figure 7C* and *Figure 7F*). This illustrates the value of predicting fluorescence from label-free contrasts: while the label-free contrasts are predictive of FluoroMyelin stain, the complex relations between them makes estimation of myelin level from label-free contrasts challenging. The neural network can learn the complex transformation from label-free contrasts to FluoroMyelin stain and enables reliable estimation of myelin levels.

## Data normalization

In addition to architecture, it is essential to devise proper image normalization for correctly predicting the intensity across different fields of view in large stitched images. We found that per-image normalization commonly applied to image segmentation tasks did not preserve the intensity variation across images and led to artifacts in prediction. The two main issues that need to be accounted for in image translation tasks are: (1) numbers of background pixels vary across images and can bias the normalization parameters if not excluded from normalization (*Yang et al., 2019*), and (2) there are batch variations in the staining and imaging process when pooling multiple datasets together for training. While batch variation is less pronounced in quantitative label-free imaging, it remains quite significant in fluorescence images of stained samples and therefore needs to be corrected. We found that normalizing per-dataset with the median of inter-quartile range of foreground pixel intensities gives the most accurate intensity prediction (*Figure 7—figure supplement 1*).

Notably, the 2D model with phase, retardance, and orientation as the input has correlation and similarity scores close to the best 2.5D model but the training takes just 3.7 hr to converge, while the best 2.5D model takes 64.7 hr to converge (*Table 4*). This is likely because the 2D phase reconstruction captures the density variation encoded in the brightfield Z-stack that is informative for the model to predict axon tracts accurately.

## Rescue of inconsistent label

Robust fluorescent labeling usually requires optimization of labeling protocols and precise control of labeling conditions. Sub-optimal staining protocols often lead to staining artifacts and make the

samples unusable. Quantitative label-free imaging, on the other hand, provides more robust measurements as it generates contrast in physical units and does not require labeling. Therefore, fluorescence images predicted from quantitative label-free inputs are more robust to experimental variations. For example, we found FluoroMyelin stain intensity faded unevenly over time and formed dark patches in the images (indicated by cyan arrow heads in *Figure 7A and D*), possibly due to quenching of FluoroMyelin by the antifade chemical in the mounting media. However, this quenching of dye does not affect the physical properties measured by the label-free channels. Therefore, the model trained on images without artifacts predicted the expected staining pattern even with the failure of experimental stain. This robustness is particularly valuable for precious tissue specimens such as archival prenatal human brain tissue.

## Discussion

We have reported QLIPP, a novel computational imaging method for label-free measurement of density and anisotropy from 3D polarization-resolved acquisition. While quantitative fluorescence imaging provides molecular specificity, quantitative label-free imaging provides physical specificity. We show that several organelles can be identified from their density and anisotropy. We also show that multiple regions of mouse brain tissue and archival human brain tissue can be identified without label. We have also reported multi-channel 2.5D U-Net deep learning architecture and training strategies to translate this physical description of the specimen to the molecular description. Next, we discuss how we elected to balance the trade-offs and the future directions of research enabled by innovations reported here.

We have designed QLIPP to be easy to adopt and multiplex with fluorescence microscopy. Using QLIPP requires a single liquid-crystal polarization modulator and a motorized Z stage. Our open-source Python software is free to use for non-profit research. Shribak (*Shribak et al., 2008*) has reported joint imaging of 2D phase and retardance with orientation-independent differential interference contrast (OI-DIC) and orientation-independent PolScope (OI-POL), which required six polarization modulators and acquisition protocol more complex than QLIPP. Ptychography-based phase retrieval method has been extended with polarization sensitive components for joint imaging of 2D phase and retardance (*Ferrand et al., 2018*; *Baroni et al., 2020*), albeit requiring hundreds of images. Our method uses one polarization modulator, compared to six used by OI-DIC, and fewer images (5 × number of Z slices), compared to hundreds in ptychography-based method, for recovery of 3D phase, retardance and orientation. Our measurements also achieve diffraction-limited resolution and provide adequate time resolution for live-cell imaging, as demonstrated by the 3D movie of U2OS cells (*Figure 2*, *Figure 2—video 2*, *Figure 2—video 3*). We anticipate that the modularity of the optical path and the availability of reconstruction software will facilitate adoption of QLIPP.

Phase information is inherently present in polarization-resolved acquisition, but can now be reconstructed using forward models and corresponding inverse algorithms reported here. We note that our approach of recovering phase from propagation of light reports the local phase variation rather than the absolute phase. Local phase variation is less sensitive to low spatial frequency or large-scale variations in density as can be seen from phase images n *Figure 2—figure supplement 3* and *Figure 6—figure supplement 1*. Recovering density at low spatial frequency requires a more elaborate optical path for creating interference with a reference beam and is more difficult to implement than QLIPP (*Kim et al., 2018*; *Popescu et al., 2006*). Nonetheless, most biological processes can be visualized with the local density variation. Further, our method uses partially coherent illumination, that is, simultaneous illumination from multiple angles, which improves spatial resolution, depth sectioning, and robustness to imperfections in the light path away from the focal plane.

QLIPP belongs to the class of polarization-resolved imaging in which the specimen is illuminated in transmission. Two other major classes of polarization-sensitive imaging are polarization sensitive optical coherence tomography (PS-OCT) and fluorescence polarization. PS-OCT is a label-free imaging method in which specimen is illuminated in reflection mode. PS-OCT has been used to measure round-trip retardance and diattenuaton of diverse tissues, for example of brain tissue (*Wang et al., 2018*). But, determination of the slow axis in the reflection mode remains challenging due to the fact that light passes through the specimen in two directions. Fluorescence polarization imaging relies on rotationally constrained fluorescent probes (*DeMay et al., 2011*; *Mehta et al., 2016*). Fluorescence

polarization measurements report the rotational diffusion and angular distribution of labeled molecules, which differs from QLIPP we have reported here.

We also note that, similar to other polarization-resolved imaging systems (*Mehta et al., 2016*), our approach reports projection of the anisotropy onto the focal plane rather than 3D anisotropy. Anisotropic structures such as axon bundles, appear isotropic to the imaging system when they are aligned along the optical axis of the imaging path. Methods for imaging 3D anisotropy with various models and systems (*Oldenbourg, 2008*; *Spiesz et al., 2011*; *Axer et al., 2011c*; *Zilles et al., 2016*; *Schmitz et al., 2018a*; *Schmitz et al., 2018b*; *Yang et al., 2018*; *Tran and Oldenbourg, 2018*) are now in active development. Recovering 3D anisotropy along with 3D density using forward models that account for diffraction effects in the propagation of polarized light would be an important area of research for the future.

We demonstrated the potential of QLIPP for sensitive detection of orientation of axon bundles (*Figure 5* and *Figure 6*). Combining these measurements with tractography algorithms can facilitate analysis of mesoscale connectivity. Tractography algorithms developed for diffusion weighted-MRI measurements (*Zhan et al., 2015*) have been adapted to brain images from a lower-resolution polarization microscope (~60 µm) (*Axer et al., 2011c*). We envision that combining tractography algorithms with anisotropy measured at optical resolution, which reports the orientation of ensemble of axons, will enable development of probabilistic models of connectivity. Although multiple methods for tracing connectivity in the mouse brain at mesoscale (cellular level) have been developed (*Ragan et al., 2012*; *Oh et al., 2014*; *Zeng, 2018*), they have not yet been extended to human brain. The volume of fetal human brain during third trimester $(10^5 \mathrm{mm}^3 - 4 \times 10^5 \mathrm{mm}^3)$ is 3 orders of magnitude larger than the volume of an adult mouse brain (~1.5 $\times$ 10$^2$ mm$^3$). Our data show that label-free measurement of myelination and axon tract orientation is possible with ~1.5 µm diffraction-limited resolution over the scale of whole fetal human brain sections. Further work in streamlining sample preparation, imaging, data curating, and model training would be required to apply QLIPP to large scale organs.

Our multi-channel 2.5D deep learning models are designed for efficient analysis of multi-dimensional 3D data. In contrast to earlier work on image translation that demonstrated 2D prediction (*Christiansen et al., 2018*; *Rivenson et al., 2018a*; *Rivenson et al., 2018b*), our 2.5D architecture is inspired by *Han, 2017* and provides comparable prediction accuracy at a lower computational cost than using 3D U-Net. Pearson correlation coefficient in 3D for nuclei prediction from brightfield images is 0.87 vs. ~0.7 reported in *Ounkomol et al., 2018*. In comparison to Christiansen et al.'s 2D translation model (*Christiansen et al., 2018*) where the image translation was formulated as a pixel-wise classification task of 8-bit classes, our 2.5D model formulates the image translation as a regression task that allows prediction of much larger dynamic range of gray levels. While training a single model that predicts multiple structures seems appealing, this more complex task requires increasing the model size with the trade-off of longer model training time. Our modeling strategy to train one model to predict only one target allowed us to use significantly smaller models that can fit into the memory of a single GPU for faster training.

We systematically evaluated how the dimensions and input channels affect the prediction accuracy. Compared to previous work that predict fluorescence images from single label-free contrast (*Ounkomol et al., 2018*; *Christiansen et al., 2018*; *Rivenson et al., 2018a*; *Rivenson et al., 2018b*), we show that higher prediction accuracy can be achieved by combining multiple label-free contrasts. Additionally, we report the image normalization strategy required to predict large images stitched from smaller fields of view from multi-channel inputs.

The image quality metrics we use to evaluate the model performance depends on the accuracy of the prediction but also the noise in the target images. A more direct comparison of model performances on the same dataset would be useful in the future. Further, the more flexible 2.5D network allows for application to image data that has only a few Z-slices without up- or down-sampling the data, making it useful for analysis of microscopic images that often has variable number of Z-slices. Even though we focus on image translation in this work, the same 2.5D network can be used for 3D segmentation. 3D segmentation using the 2.5D network bears additional advantages over 3D network, because sparse annotation can be done on a subset of slices sampled from the 3D volume, while 3D network requires all the slices in the input volume to be annotated. The flexibility of sparse annotation allows for better sampling of structural variation in the data with the same effort on manual annotation.

A common shortfall of machine learning approaches is that they tend not to generalize well. We have shown that our data normalization and training process leads to models of myelination that generalize to two developmental time points. In contrast to reconstruction using physical models, the errors or artifacts in the prediction by machine learning models are highly dependant on the quality of training data and their similarity to the new input data. Therefore, prediction errors made by the machine learning models are difficult to identify in the absence of ground truth. Extending the image translation models such that they predict not just the value, but also provide estimate of the confidence interval of output values, is an important area of research.

## Conclusion

In summary, we report reconstruction of specimen density and anisotropy using quantitative label-free imaging with phase and polarization (QLIPP) and prediction of fluorescence distribution from label-free images using deep convolutional neural networks. Our reconstruction algorithms (https://github.com/mehta-lab/reconstruct-order) and computationally efficient U-Net variants (https://github.com/czbiohub/microDL) facilitate measurement and interpretation of physical properties of the specimens. We reported joint measurement of phase, retardance, and orientation with diffraction-limited spatial resolution in 3D dividing cells and in 2D brain tissue slices. We demonstrated visualization of diverse biological structures: axon tracts and myelination in mouse and human brain slices, and multiple organelles in cells. We demonstrated accurate prediction of fluorescent images from density and anisotropy with multi-contrast 2.5D U-Net model. We demonstrated strategies for accurate prediction myelination in centimeter-scale prenatal human brain tissue slices. We showed that inconsistent labeling of human tissue can be rescued with qualitative label-free imaging and trained models. We anticipate that our approach will enable quantitative label-free analysis of architectural order at multiple spatial and temporal scales, particularly in live cells and clinically-relevant tissues.

# Materials and methods

**Key resources table**

| Reagent type (species) or resource | Designation | Source or reference | Identifiers | Additional information |
|---|---|---|---|---|
| biological sample (*M. musculus*) | mouse kidney tissue section | Thermo-Fisher Scientific | Cat. # F24630 | |
| biological sample (*M. musculus*) | mouse brain tissue section | this paper | | mouse line maintained in M. Han lab, see *Specimen preparation* in Materials and methods |
| biological sample (*H. sapiens*) | developing human brain tissue section | this paper | | archival tissue stored in T. Nowakowski lab, see *Specimen preparation* in Materials and methods |
| chemical compound, drug | FluoroMyelin | Thermo-Fisher Scientific | Cat. # F34652 | |
| software, algorithm | reconstruction algorithms | https://github.com/mehta-lab/reconstruct-order | | |
| software, algorithm | 2.5 U-Net | https://github.com/czbiohub/microDL | | |
| software, algorithm | Micro-Manager 1.4.22 | https://micro-manager.org/ | RRID:SCR_016865 | |
| software, algorithm | OpenPolScope | https://openpolscope.org/ | | |

## Model of image formation

We describe dependence of the polarization resolved images on the specimen properties using Stokes vector representation of partially polarized light (*Bass et al., 2009*, Ch.15). This representation allows us to accurately measure the polarization sensitive contrast in the imaging volume. First, we retrieve the coefficients of the specimen's Mueller matrix that report linear retardance, slow-axis orientation, transmission (brightfield), and degree of polarization. For brevity, we call them 'Mueller

coefficients' of the specimen in this paper. Mueller coefficients are recovered from the polarization-resolved intensities using the inverse of an instrument matrix that captures how Mueller coefficients are related to acquired intensities. Assuming that the specimen is mostly transparent, more specifically satisfies the first Born approximation (*Born and Wolf, 2013*), we reconstruct specimen phase, retardance, slow axis, and degree of polarization stacks from stacks of Mueller coefficients. The assumption of transparency is generally valid for the structures we are interested in, but does not necessarily hold when the specimen exhibits significant absorption or diattenuation. To ensure that the inverse computation is robust, we need to make judicious decisions about the light path, calibration procedure, and background estimation. A key advantage of Stokes instrument matrix approach is that it easily generalizes to other polarization diverse imaging methods - A polarized light microscope is represented directly by a calibrated instrument matrix.

For sensitive detection of retardance, it is advantageous to suppress isotropic background by illuminating the specimen with elliptically polarized light of opposite handedness to the analyzer in the detector side (*Shribak and Oldenbourg, 2003*). For experiments reported in this paper, we acquired data by illuminating the specimen sequentially with right-handed circular and elliptical polarized light and analyzed the transmitted left-handed circular polarized light in detection.

### Forward model: specimen properties → Mueller coefficients

We assume a weakly scattering specimen modeled by properties of linear retardance $\rho$, orientation of the slow axis $\omega$, transmission $t$, and depolarization $p$. The Mueller matrix of the specimen can be expressed as a product of two Mueller matrices, $\mathbf{M}_t$, accounting for the effect of transmission and depolarization from the specimen, and $\mathbf{M}_r$, accounting for the effect of retardance and orientation of the specimen. The expression of $\mathbf{M}_r$ is a standard Mueller matrix of a linear retarder that can be found in *Bass et al., 2009*, Ch.14, and $\mathbf{M}_t$ is expressed as

$$\mathbf{M}_t = \begin{bmatrix} t & 0 & 0 & 0 \\ 0 & tp & 0 & 0 \\ 0 & 0 & tp & 0 \\ 0 & 0 & 0 & tp \end{bmatrix}. \tag{1}$$

With $\mathbf{M}_t$ and $\mathbf{M}_r$, the Mueller matrix of the specimen is then given by

$$\mathbf{M}_{sm} = \mathbf{M}_t \cdot \mathbf{M}_r = \begin{bmatrix} m_0 & 0 & 0 & 0 \\ 0 & * & * & m_1 \\ 0 & * & * & m_2 \\ 0 & -m_1 & -m_2 & m_3 \end{bmatrix}, \tag{2}$$

where * signs denote irrelevant entries that cannot be retrieved under our experiment scheme. The relevant entries that are retrievable can be expressed as a vector of Mueller coefficients, which is

$$\mathbf{m} = \begin{bmatrix} m_0 \\ m_1 \\ m_2 \\ m_3 \end{bmatrix} = \begin{bmatrix} t \\ tp \sin 2\omega \sin \rho \\ -tp \cos 2\omega \sin \rho \\ tp \cos \rho \end{bmatrix} \tag{3}$$

This vector is coincidentally the Stokes vector when right-handed circularly polarized light passing through the specimen. The aim of the measurement we describe in the following paragraphs is to accurately measure these Mueller coefficients at each point in the image plane of the microscope by illuminating the specimen and detecting the scattered light with mutually independent polarization states. Once a map of these Mueller coefficients has been acquired with high accuracy, the specimen properties can be retrieved from the above set of equations.

### Forward model: Mueller coefficients → intensities

To acquire the above Mueller coefficients, we illuminate the specimen with a series of right-handed circularly and elliptically polarized light (*Shribak and Oldenbourg, 2003*). The Stokes vectors of our sequential illumination states are given by,

$$\mathbf{S}_i = \begin{bmatrix} 1 \\ 0 \\ 0 \\ 1 \end{bmatrix}_{i=\mathrm{RCP}}, \begin{bmatrix} 1 \\ \sin\chi \\ 0 \\ \cos\chi \end{bmatrix}_{i=0}, \begin{bmatrix} 1 \\ -\sin\chi \\ 0 \\ \cos\chi \end{bmatrix}_{i=45},$$

$$\begin{bmatrix} 1 \\ 0 \\ \sin\chi \\ \cos\chi \end{bmatrix}_{i=90}, \begin{bmatrix} 1 \\ 0 \\ -\sin\chi \\ \cos\chi \end{bmatrix}_{i=135} \tag{4}$$

where $\chi$ is the compensatory retardance controlled by the LC that determines the ellipticity of the four elliptical polarization states.

After our controlled polarized illumination has passed through the specimen, we detect the left-handed circular polarized light by having a left-handed circular analyzer in front of our sensor. We express the Stokes vector before the sensor as

$$\mathbf{S}_{\mathrm{sensor},i} = \mathbf{M}_{\mathrm{LCA}}\mathbf{M}_{\mathrm{sm}}\mathbf{S}_i, \tag{5}$$

where $i = \{\mathrm{RCP}, 0, 45, 90, 135\}$ depending on the illumination states, and $\mathbf{M}_{\mathrm{LCA}}$ is the Muller matrix of a left-handed circular analyzer (*Bass et al., 2009*, Ch.14). The detected intensity images are the first component of Stokes vector at the sensor under different illuminations ($I_i = [\mathbf{S}_{\mathrm{sensor},i}]_0$). Stacking the measured intensity images to form a vector

$$\mathbf{I} = \begin{bmatrix} I_{\mathrm{RCP}} \\ I_0 \\ I_{45} \\ I_{90} \\ I_{135} \end{bmatrix}, \tag{6}$$

we can link the relationship between the measured intensity and the specimen vector through an 'instrument matrix' $\mathbf{A}$ as

$$\mathbf{I} = \mathbf{A}\mathbf{m}, \tag{7}$$

where

$$\mathbf{A} = \begin{bmatrix} 1 & 0 & 0 & -1 \\ 1 & \sin\chi & 0 & -\cos\chi \\ 1 & 0 & \sin\chi & -\cos\chi \\ 1 & -\sin\chi & 0 & -\cos\chi \\ 1 & 0 & -\sin\chi & -\cos\chi \end{bmatrix}. \tag{8}$$

Each row of the instrument matrix is determined by the interaction between various illumination polarization states and the specimen's properties. Any polarization-resolved measurement scheme can be characterized by an instrument matrix that transforms specimen's polarization property to the measured intensities. Calibration of the polarization imaging system is then done through calibrating this instrument matrix.

## Computation of Mueller coefficients at image plane

Once the instrument matrix has been experimentally calibrated, the vector of Mueller coefficients can be obtained from recorded intensities using its inverse (compare *Equation 7*),

$$\mathbf{m} = \mathbf{A}^{-1}\mathbf{I}, \tag{9}$$

## Computation of background corrected specimen properties

We retrieved the vector of Mueller coefficients, $\mathbf{m}$, by solving *Equation 9*. Slight strain or misalignment in the optical components or the specimen chamber can lead to background that masks out

contrast from the specimen. The background typically varies slowly across the field of view and can introduce spurious correlations in the measurement. It is crucial to correct the vector of Mueller coefficients for non-uniform background retardance that was not accounted for by the calibration process. To correct the non-uniform background retardance, we acquired background polarization images at the empty region of the specimen. We then transformed specimen ($i = \mathrm{sm}$) and background ($i = \mathrm{bg}$) vectors of Mueller coefficients as follows,

$$\begin{aligned}
\overline{m_1}^i &= m_1^i / m_3^i, \\
\overline{m_2}^i &= m_2^i / m_3^i, \\
\mathrm{DOP}^i &= \frac{\sqrt{(m_1^i)^2 + (m_2^i)^2 + (m_3^i)^2}}{m_0^i},
\end{aligned} \tag{10}$$

We then reconstructed the background corrected properties of the specimen: brightfield (BF), retardance (ρ), slow axis (ω), and degree of polarization (DOP) from the transformed specimen and background vectors of Mueller coefficients $\overline{\mathbf{m}}^{\mathrm{sm}}$ and $\overline{\mathbf{m}}^{\mathrm{bg}}$ using the following equations:

$$\overline{m_1} = \overline{m_1}^{\mathrm{sm}} - \overline{m_1}^{\mathrm{bg}} \tag{11}$$

$$\overline{m_2} = \overline{m_2}^{\mathrm{sm}} - \overline{m_2}^{\mathrm{bg}} \tag{12}$$

$$\mathrm{BF} = m_0^{\mathrm{sm}} / m_0^{\mathrm{bg}} \tag{13}$$

$$\rho = \arctan 2 \left( \sqrt{\overline{m_1}^2 + \overline{m_2}^2} \right) \tag{14}$$

$$\omega = \frac{1}{2} \arctan 2 \left( \frac{\overline{m_1}}{-\overline{m_2}} \right) \tag{15}$$

$$\mathrm{DOP} = \mathrm{DOP}^{\mathrm{sm}} / \mathrm{DOP}^{\mathrm{bg}} \tag{16}$$

When the background cannot be completely removed using the above background correction strategy with a single background measurement, (i.e. the specimen has spatially varying background retardance), we applied a second round of background correction on the measurements. In this second round, we estimated the residual transformed background Mueller coefficients by fitting a low-order 2D polynomial surface to the transformed specimen Mueller coefficients. Specifically, we downsampled each 2048 × 2048 image to 64 × 64 image with 32 × 32 binning. We took the median of each 32 × 32 bin to be each pixel value in the downsampled image. We then fitted a second-order 2D polynomial surface to the downsampled image of each transformed specimen Mueller coefficient to estimate the residual background. With this newly estimated background, we performed another background correction. The effects of two rounds of the background corrections are shown in *Figure 2—figure supplement 2*.

## Phase reconstruction

As seen from *Equation 3*, the first component in the vector of Mueller coefficients, $m_0$, is equal to the total transmitted intensity of electric field in the focal plane. Assuming a specimen with weak absorption, the intensity variations in a Z-stack encode the phase information via the transport of intensity (TIE) equation (*Streibl, 1984*). In the following, we leverage weak object transfer function (WOTF) formalism (*Streibl, 1985*; *Noda et al., 1990*; *Claus et al., 2015*; *Jenkins and Gaylord, 2015a*; *Jenkins and Gaylord, 2015b*; *Soto et al., 2017*) to retrieve 2D and 3D phase from this TIE phase contrast and describe the corresponding inverse algorithm.

### Forward model for phase reconstruction

The linear relationship between the 3D phase and the through focus brightfield intensity was established in *Streibl, 1985* with Born approximation and weak object approximation. In our context, we reformulated as (*Streibl, 1985*; *Noda et al., 1990*; *Soto et al., 2017*)

$$m_0(\mathbf{r}) = m_{0,\mathrm{dc}} + \phi(\mathbf{r}) \otimes_\mathbf{r} h_\phi(\mathbf{r}) + \mu(\mathbf{r}) \otimes_\mathbf{r} h_\mu(\mathbf{r}), \tag{17}$$

where $\mathbf{r} = (\mathbf{r}_\perp, z) = (x, y, z)$ is the 3D spatial coordinate vector, $m_{0,\mathrm{dc}}$ is the constant background of $m_0$ component, $\otimes_\mathbf{r}$ denotes convolution operation over $\mathbf{r}$ coordinate, Φ refers to phase, μ refers to absorption, $h_\phi(\mathbf{r})$ is the phase point spread function (PSF), and $h_\mu(\mathbf{r})$ is the absorption PSF. Strictly,

$\Phi$ and $\mu$ are the real and imaginary part of the scattering potential scaled by $\Delta z/2k$, where $\Delta z$ is the axial pixel size of the experiment and $k$ is the wavenumber of the incident light. When the refractive index of the specimen and that of the environment are close, the real and imaginary scaled scattering potential reduce to two real quantity, phase and absorption.

When specimen's thickness is larger than the depth of field of the microscope (usually in experiments with high NA objective), the brightfield intensity stack contains 3D information of specimen's phase and absorption. Without making more assumptions or taking more data, solving 3D phase and absorption from 3D brightfield is ill-posed because we are solving two unknowns from one measurement. Assuming the absorption of the specimen is negligible (*Noda et al., 1990*; *Jenkins and Gaylord, 2015b*; *Soto et al., 2017*), which generally applies to transparent biological specimens, we turn this problem into a linear deconvolution problem, where 3D phase is retrieved.

When specimen's thickness is smaller than the depth of field of the microscope (usually in experiments with low NA objective), the whole 3D intensity stack is coming from merely one effective 2D absorption and phase layer of specimen. We rewrite *Equation 17* as (*Claus et al., 2015*; *Jenkins and Gaylord, 2015a*)

$$m_0(\mathbf{r}) = m_{0,\text{dc}} \;+\; \phi(\mathbf{r}_\perp) \otimes_{\mathbf{r}_\perp} h_\phi(\mathbf{r}_\perp, z) \\ +\; \mu(\mathbf{r}_\perp) \otimes_{\mathbf{r}_\perp} h_\mu(\mathbf{r}_\perp, z). \tag{18}$$

In this situation, we have multiple 2D defocused measurements to solve for one layer of 2D absorption and phase of the specimen.

## Inverse problem for phase reconstruction

With the linear relationship between the first component of the Mueller coefficients vector and the phase, we then formulated the inverse problem to retrieve 2D and 3D phase of the specimen.

When we recognize the specimen as a 3D specimen, we then use *Equation 17* and drop the absorption term to estimate specimen's 3D phase through the following optimization algorithm:

$$\min_{\phi(\mathbf{r})} \sum_{\mathbf{r}} \left| m_0'(\mathbf{r}) - \phi(\mathbf{r}) \otimes_{\mathbf{r}} h_\phi(\mathbf{r}) \right|^2 + \tau_\phi \text{Reg}(\phi(\mathbf{r})), \tag{19}$$

where $m_0'(\mathbf{r}) = m_0(\mathbf{r}) - m_{0,\text{dc}}$, $\tau_\phi$ is the regularization parameter for applying different degree of denoising effect, and the regularization term depending on the choice of either Tikhonov or anisotropic total variation (TV) denoiser is expressed

$$\text{Reg}(\phi(\mathbf{r})) = \begin{cases} \displaystyle\sum_{\mathbf{r}} |\phi(\mathbf{r})|^2, & \text{Tikhonov} \\ \displaystyle\sum_{\mathbf{r}} \sum_{i=x,y,z} |\partial_i \phi(\mathbf{r})|, & \text{TV} \end{cases}$$

When using Tikhonov regularization, this optimization problem has an analytic solution that has previously described by *Noda et al., 1990*; *Jenkins and Gaylord, 2015b*; *Soto et al., 2017*. As for TV regularization, we adopted alternating minimization algorithm that is proposed and applied to phase imaging in *Wang et al., 2008* and *Chen et al., 2018*, respectively, to solve the problem.

If we consider the specimen as a 2D specimen, we then turn *Equation 18* into the following optimization problem:

$$\min_{\phi, \mu(\mathbf{r}_\perp)} \sum_{\mathbf{r}} \left| m_0'(\mathbf{r}) - \phi(\mathbf{r}_\perp) \otimes_{\mathbf{r}_\perp} h_\phi(\mathbf{r}_\perp, z) - \mu(\mathbf{r}_\perp) \otimes_{\mathbf{r}_\perp} \right. \\ \left. h_\mu(\mathbf{r}_\perp, z) \right|^2 + \tau_\phi \text{Reg}(\phi(\mathbf{r}_\perp)) + \tau_\mu \text{Reg}(\mu(\mathbf{r}_\perp)), \tag{20}$$

where we have an extra regularization parameter $\tau_\mu$ here for the absorption. When Tikhonov regularization is selected, the analytic solution similar to the one described in *Chen et al., 2016* is adopted.

When the signal to noise ratio of the brightfield stack is high, Tikhonov regularization gives satisfactory reconstruction in a single step with computation time proportional to the size of the image stack. However, when the noise is high, Tikhonov regularization can lead to high- to medium-frequency artifacts. Using iterative TV denoising algorithm, we can trade-off reconstruction speed with robustness to noise.

## Specimen preparation

Mouse kidney tissue slices were purchased (Thermo-Fisher Scientific). In the mouse kidney tissue slice, F-actin was labeled with Alexa Fluor 568 phalloidin and nuclei was labeled with DAPI. U2OS cells were seeded and cultured in a chamber made of two strain-free coverslips that allowed for gas exchange.

## Mouse brain section

The mice were anesthetized by inhalation of isoflurane in a chemical fume hood and then perfused with 25 ml phosphate-buffered saline (PBS) into the left cardiac ventricle and subsequently with 25 ml of 4% paraformaldehyde (PFA) in the PBS solution. Thereafter, the brains were post-fixed with 4% PFA for 12–16 hr and then transferred to 30% sucrose solution at the temperature of 4°C for 2–3 days until the tissue sank to the bottom of the container. Then, the brains were embedded in a tissue freezing medium (Tissue-Tek O.C.T compound 4583, Sakura) and kept at the temperature of −80°C. Cryostat-microtome (Leica CM 1850, Huston TX) was used for preparing the tissue sections (12 and 50 μm) at the temperature of −20°C and the slides were stored at the temperature of −20°C until use. In order to analyze myelination with QLIPP, the OCT on the slides were melted by keeping the slides at 37°C for 15–30 min. Then, the slides were washed in PBST (PBS+Tween-20 [0.1%]) for five minutes and then washed in PBS for five minutes and coversliped by mounting media (F4680, FluromountTM aqueousm sigma).

## Prenatal human brain section

De-identified brain tissue samples were received with patient consent in accordance with a protocol number approved by the Human Gamete, Embryo, and Stem Cell Research Committee (institutional review board) at the University of California, San Francisco. Human prenatal brain samples were fixed with 4% paraformaldehye in phosphate-buffered solution (PBS) overnight, then rinsed with PBS, dehydrated in 30% sucrose/OCT compound (Agar Scientific) at 4°C overnight, then frozen in OCT at −80°C. Frozen samples were sectioned at 12 $\mu m$ and mounted on microscope slides. Sections were stained directly with red FluoroMyelin (Thermo-Fisher Scientific, 1:300 in PBS) for 20 min at room temperature, rinsed three times with PBS for 10 min each, then mounted with ProLong Gold antifade (Invitrogen) with a coverslip.

## Image acquisition and registration

We implemented LC-PolScope on a Leica DMi8 inverted microscope with Andor Dragonfly confocal for multiplexed acquisition of polarization-resolved images and fluorescence images. We automated the acquisition using Micro-Manager v1.4.22 and OpenPolScope plugin for Micro-Manager that controls liquid crystal universal polarizer (custom device from Meadowlark Optics, specifications available upon request).

We multiplexed the acquisition of label-free and fluorescence volumes. The volumes were registered using transformation matrices computed from similarly acquired multiplexed volumes of 3D matrix of rings from the ARGO-SIM test target (Argolight).

In transmitted light microscope, the resolution increases and image contrast decreases with increased numerical aperture of illumination. We used 63 × 1.47 NA oil immersion objective (Leica) and 0.9 NA condenser to achieve a good balance between image contrast and resolution. The mouse kidney tissue slice was imaged using 100 ms exposure for five polarization channels, 200 ms exposure for 405 nm channel (nuclei) at 1.6 mW in the confocal mode, 100 ms exposure for 561 nm channel (F-actin) at 2.8 mW in the confocal mode. The mouse brain slice were imaged using 30 ms exposure for five polarization channels. U2OS cells were imaged using 50 ms exposure for five polarization channels. For training the neural network, we acquired 160 non-overlapping 2048 × 2048 × 45 z-stacks of the mouse kidney tissue slice with Nyquist sampled voxel size 103 nm × 103 nm ×250 nm. Human brain sections were imaged with a 10 × 0.3 NA objective and 0.2 NA condenser with a 200 ms exposure for polarization channels, 250 ms exposure for 568 channel (FluoroMyelin) in the epifluorescence mode. The full brain sections were imaged, approximately 200 images depending on the size of the section, with 5 Z-slices at each location. The registered images mouse kidney tissue slice are available in the BioImage Archive (https://www.ebi.ac.uk/biostudies/BioImages/studies/S-BIAD25).

## Data preprocessing for model training

The images were flat-field corrected. For training 3D models, the image volumes were upsampled along Z to match the pixel size in XY using linear interpolation. The images were tiled into $256 \times 256$ patches with a 50% overlap between patches for 2D and 2.5D models. The volumes were tiled into $128 \times 128 \times 96$ patches for 3D models with a 25% overlap along XYZ. Tiles that had sufficient fluorescence foreground (2D and 2.5D: 20%, 3D: 50%) were used for training. Foreground masks were computed by summing the binary images of nuclei and F-actin obtained from Otsu thresholding in the case of mouse kidney tissue sections, and binary images of FluoroMyelin for the human brain sections. Images of human brain sections were visually inspected and curated to exclude images containing quenching artifacts as shown in *Figure 7* before training.

Proper data normalization is essential for predicting the intensity correctly across different fields-of-views. We found the common normalization scheme where each image is normalized by its mean and standard deviation does not produce correct intensity prediction (*Figure 7—figure supplement 1*). We normalized the images on a per dataset basis to correct the batch variation in the staining and imaging process across different datasets. To balance contributions from different channels during training of multi-contrast models, each channel needs to be scaled to similar range. Specifically, for each channel, we subtracted its median and divided by its inter-quartile range (range defined by 25% and 75% quantiles) of the foreground pixel intensities. We used inter-quartile range to normalize the channel because standard deviation underestimates the spread of the distribution of highly correlated data such as pixels in images.

## Neural network architecture

We experimented with 2D, 2.5D and 3D versions of U-Net models *Figure 3—figure supplement 1*. Across the three U-Net variants, each convolution block in the encoding path consists of two repeats of three layers: a convolution layer, ReLU non-linearity activation, and a batch normalization layer. We added a residual connection from the input of the block to the output of the block to facilitate faster convergence of the model (*Milletari et al., 2016*; *Drozdzal et al., 2016*). $2 \times 2$ downsampling is applied with $2 \times 2$ convolution with stride two at the end the each encoding block. On the decoding path, the feature maps were passed through similar convolution blocks, followed by up-sampling using bilinear interpolation. Feature maps output by every level of encoding path were concatenated to feature maps in the decoding path at corresponding levels. The final output block had a convolution layer only.

The encoding path of our 2D and 2.5D U-Net consists of five layers with 16, 32, 64, 128 and 256 filters respectively, while the 3D U-Net consists of four layers with 16, 32, 64 and 128 filters each due to its higher memory requirement. The 2D and 3D versions use convolution filters of size of $3 \times 3$ and $3 \times 3 \times 3$ with a stride of 1 for feature extraction and with a stride of 2 for downsampling between convolution blocks.

The 2.5D U-Net has the similar architecture as the 2D U-Net with following differences:

1. The 3D features maps are converted into 2D using skip connections that consist of a $N \times 1 \times 1$ valid convolution, where $N = 3, 5, 7$ is the number of slices in the input.
2. Convolution filters in the encoding path are $N \times 3 \times 3$.
3. In the encoding path, the feature maps are downsampled across blocks using $N \times 2 \times 2$ average pooling.
4. In the decoding path, the feature maps were upsampled using bilinear interpolation by a factor of $1 \times 2 \times 2$ and the convolution filters in the decoding path are of shape $1 \times 3 \times 3$.

The 2D , 2.5D, 3D network with single channel input consisted of 2.0 M, 4.8M, 1.5M learnable parameters, respectively.

## Model training and inference

We randomly split the images in groups of 70%, 15%, and 15% for training, validation and test. The split are kept consistent across all model training to make the results comparable. All models are trained with Adam optimizer, L1 loss function, and a cyclic learning rate scheduler with a min and max learning rate of $5 \times 10^{-5}$ and $6 \times 10^{-3}$ respectively. The 2D, 2.5D, 3D network were trained on mini-batches of size 64, 16, and four to accommodate the memory requirements of each model. Models were trained until there was no decrease in validation loss for 20 epochs. The model with

minimal validation loss was saved. Single channel 2D models converged in 6 hr, 2.5D model converged in 47 hr and the 3D model converged in 76 hr on NVIDIA Tesla V100 GPU with 32 GB RAM.

As the models are fully convolutional, model predictions were obtained using full XY images as input for the 2D and 2.5D versions. Due to memory requirements of the 3D model, the test volumes were tiled along x and y while retaining the entire z extent (patch size: $512 \times 512 \times 96$) with an overlap of 32 pixels along X and Y. The predictions were stitched together by linear blending of the model predictions across the overlapping regions. Inference time for a single channel U-Net model was 105, 3 and 18 seconds/frame for 2D, 2.5D, and 3D models respectively, with $2048 \times 2048$ pixels to a frame.

## Model evaluation

Pearson correlation and structural similarity index (SSIM) along the XY, XZ and XYZ dimensions of the test volumes were used for evaluating model performance.

The Pearson correlation coefficient between a target image $T$ and a prediction image $P$ is defined as

$$r(T,P) = \frac{\sigma_{TP}}{\sigma_T \sigma_P} \tag{21}$$

where $\sigma_{TP}$ is the covariance of $T$ and $P$, and $\sigma_T$ and $\sigma_P$ are the standard deviations of $T$ and $P$ respectively.

SSIM compares two images using a sliding window approach, with window size $N \times N$ ($N \times N \times N$ for XYZ). Assuming a target window $t$ and a prediction window $p$,

$$SSIM(t,p) = \frac{\left(2\mu_t\mu_p + c_1\right)\left(2\sigma_{tp} + c_2\right)}{\left(\mu_t^2 + \mu_p^2 + c_1\right)\left(\sigma_t^2 + \sigma_p^2 + c_2\right)} \tag{22}$$

where $c_1 = (0.01L)^2$ and $c_2 = (0.03L)^2$, and $L$ is the dynamic range of pixel values. Mean and variance are represented by $\mu$ and $\sigma^2$ respectively, and the covariance between $t$ and $p$ is denoted $\sigma_{tp}$. We use $N = 7$. The total SSIM score is the mean score calculated across all windows, $SSIM(T,P) = \frac{1}{M}\sum SSIM(t,p)$ for a total of $M$ windows. For XY and XZ dimensions, we compute one test metric per plane and for XYZ dimension, we compute one test metric per volume.

Importantly, it is essential to scale the the model prediction back to the original range before normalization for correct calculation of target-prediction SSIM. This is because unlike Pearson correlation coefficient, SSIM is not a scale-independent metrics.

## Acknowledgements

We thank Spyros Dermanis (CZ Biohub) and Bing Wu (CZ Biohub) for providing the mouse brain slice used for acquiring data shown in *Figure 5*. We thank Greg Huber, Loic Royer, Joshua Batson, Jim Karkanias, Joe DeRisi, and Steve Quake from the Chan Zuckerberg Biohub for numerous discussions. We also thank Eva Dyer from Georgia Tech for discussions about applications of the 2.5D models. This research was supported by the Chan Zuckerberg Biohub.

## Additional information

### Funding

| Funder | Author |
| --- | --- |
| Chan Zuckerberg Biohub | Syuan-Ming Guo<br>Li-Hao Yeh<br>Jenny Folkesson<br>Ivan E Ivanov<br>Matthew G Keefe<br>David Shin<br>Bryant B Chhun<br>Nathan H Cho<br>Tomasz J Nowakowski<br>Shalin B Mehta |

The funders had no role in study design, data collection and interpretation, or the decision to submit the work for publication.

### Author contributions

Syuan-Ming Guo, Li-Hao Yeh, Conceptualization, Resources, Data curation, Software, Formal analysis, Validation, Investigation, Visualization, Methodology, Writing - original draft, Writing - review and editing; Jenny Folkesson, Resources, Data curation, Software, Formal analysis, Supervision, Validation, Visualization, Methodology, Writing - original draft; Ivan E Ivanov, Conceptualization, Software, Validation, Investigation, Visualization, Methodology, Writing - review and editing; Anitha P Krishnan, Data curation, Software, Formal analysis, Validation, Visualization; Matthew G Keefe, Resources, Investigation, Methodology, Writing - review and editing; Ezzat Hashemi, Resources, Methodology, Writing - original draft; David Shin, Resources, Methodology; Bryant B Chhun, Resources, Data curation, Software, Validation; Nathan H Cho, Resources, Investigation; Manuel D Leonetti, Resources, Supervision; May H Han, Resources, Supervision, Writing - review and editing; Tomasz J Nowakowski, Conceptualization, Resources, Supervision, Project administration, Writing - review and editing; Shalin B Mehta, Conceptualization, Resources, Software, Formal analysis, Supervision, Funding acquisition, Validation, Investigation, Visualization, Methodology, Writing - original draft, Project administration, Writing - review and editing

### Author ORCIDs

Li-Hao Yeh (ID) http://orcid.org/0000-0003-2803-5996
Jenny Folkesson (ID) http://orcid.org/0000-0002-4673-0522
Shalin B Mehta (ID) https://orcid.org/0000-0002-2542-3582

### Ethics

Human subjects: De-identified brain tissue samples were received with patient consent in accordance with a protocol approved by the Human Gamete, Embryo, and Stem Cell Research Committee (institutional review board) at the University of California, San Francisco.

### Decision letter and Author response

Decision letter https://doi.org/10.7554/eLife.55502.sa1
Author response https://doi.org/10.7554/eLife.55502.sa2

## Additional files

### Supplementary files

• Transparent reporting form

## Data availability

Our experiments generated imaging data from mouse kidney tissue and human brain tissue slices that are useful for machine learning and other analyses. The data are available in the BioImage Archive (http://www.ebi.ac.uk/bioimage-archive) under accession number S-BIAD25.

The following dataset was generated:

| Author(s) | Year | Dataset title | Dataset URL | Database and Identifier |
|---|---|---|---|---|
| Guo SM, Yeh LH, Folkesson J, Ivanov IE, Krishnan AP, Keefe MG, Hashemi E, Shin D, Chhun B, Cho N, Leonetti M, Han MH, Nowakowski TJ, Mehta S | 2020 | Revealing architectural order with quantitative label-free imaging and deep learning | https://www.ebi.ac.uk/biostudies/BioImages/studies/S-BIAD25?query=S-BIAD25 | BioImage Archive, S-BIAD25 |

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

## Appendix 1

## Glossary

- QLIPP: Quantitative label-free imaging with phase and polarization.
- specimen phase: optical path length (OPL) of the specimen that is proportional to the product of its thickness and difference in the refractive index relative to the surrounding medium.
- specimen anisotropy: angular anisotropy in OPL, which refers to retardance and slow axis orientation collectively.
- wavefront: a surface in 3D space over which the time of propagation of light from the source is constant.
- phase of a wavefront: time delay of the wavefront, which is affected by the specimen phase.
- retardance: The difference in OPL induced by anisotropy specimen due to polarization-dependent refractive index.
- slow axis orientation: the orientation along which the refractive index of an anisotropic material is the highest. The light polarized along the slow axis experiences the highest phase delay relative to the light polarized along the other axes.
- U-Net: A fully convolution network consisting of a contracting and an expansive path, giving the architecture its U-shape.
- 2D (Slice→Slice) U-Net: A U-Net model, using $3 \times 3$ convolution filters, that predicts a 2D slice from a 2D input slice.
- 2.5D (Stack→Slice) U-Net: A U-Net model, using $N \times 3 \times 3$ convolution filters in the contracting path and $1 \times 3 \times 3$ in the expansive path, that predicts a 2D slice from a small stack of $N, (N = 3, 5, 7)$ input slices.
- 3D (Stack→Stack) U-Net: A U-Net model, using $3 \times 3 \times 3$ convolution filters, that predicts a 3D stack from a 3D input stack.
- Normalization per tile: The data used for training neural networks are split into tiles. In this normalization strategy, each time is normalized independently to have zero mean and unit variance.
- Normalization per field of view: In this normalization strategy, each field of view (which consists of 16 tiles) is normalized to have zero mean and unit variance. The variations across tiles capture variations in the input and target data over the field of view.
- Normalization per dataset: In this normalization strategy, whole training set is normalized to have zero mean and unit variance. The variations across tiles capture variations in the input and target data over the entire dataset, for example, a large brain slice.
- SSIM: The Structural SIMilarity (SSIM) index is a method for measuring the similarity between two images.

