## [Decision Letter]

**Acceptance summary:**

This paper develops a new computational imaging approach for label-free imaging that uses polarization to acquire joint measurements of density and anisotropy. The authors provide evidence that this label-free approach can be combined with deep learning to effectively characterize the architecture of different samples across multiple spatial scales. To achieve this, they introduce a relatively computationally efficient deep learning architecture based off of the 3D U-Net that can be used to predict structures from multi-channel images as well as rescue inconsistent labelling.

Overall, the topic is of high interest and the reviewers agree that combining quantitative label-free imaging and deep neural networks of live and postmortem tissue is novel and important. The work is therefore of interest to a broad scientific audience.

**Decision letter after peer review:**

Thank you for submitting your article "Revealing architectural order with quantitative label-free imaging and deep learning" for consideration by *eLife*. Your article has been reviewed by three peer reviewers, and the evaluation has been overseen by a Reviewing Editor and Vivek Malhotra as the Senior Editor. The following individual involved in review of your submission has agreed to reveal their identity: David Van Valen (Reviewer #2).

The reviewers have discussed the reviews with one another and the Reviewing Editor has drafted this decision to help you prepare a revised submission.

Summary:

This paper develops a new computational imaging approach for label-free imaging that uses polarization to acquire joint measurements of density and anisotropy. The authors provide evidence that this label-free approach can be combined with deep learning to effectively characterize the architecture of different samples across multiple spatial scales. To achieve this, they introduce a relatively computationally efficient deep learning architecture based off of the 3D U-Net that can be used to predict structures from multi-channel images as well as rescue inconsistent labelling.

Overall, the topic is rated very high and the reviewers agree that combining quantitative label-free imaging and deep neural networks of live and postmortem tissue is valued. However, there are several major concerns the authors need to address before this manuscript can be considered for publication in *eLife*. These are listed below.

Essential revisions:

1) The idea of label-free prediction of a specific tissue structure from density and anisotropy measurements is appealing and well described in the manuscript. However, the selected types of tissue and some of the general conclusions drawn (partially from cross-comparisons) are disputable. What makes mouse kidney, mouse brain and prenatal human brain unique in terms of 'revealing architectural order', but still comparable? There seems to be a lack in knowledge of brain anatomy and morphology, which is important to evaluate the results. Although the GW24 and GW20 measurements are exciting, the shown tiny ROIs are not suitable for highlighting the anatomical differences in a convincing way. The deep learning approaches appear to be correctly implemented and applied, but their scalability does not become obvious (although stated in the Abstract). A critical debate about the efforts needed to address entire large-scale organs is missing. The major add-on to state-of-the-art approaches or to previous own publications does not become clear enough.

2).Ethics: There is no clear statement of the authors concerning ethical approval, origin of samples, etc. The treatment of prenatal human tissue requires other information than the mouse brain and kidney tissue!

3) The key insight offered by this paper is that because deep learning is data-driven, these methods can be improved by improving data rather than making substantial changes to the algorithms. If there is information missing in the images that is needed to make accurate predictions, why not add it in? To me this is an under-appreciated insight, one that the authors cleverly take advantage of, and one that the life science community as a whole sorely needs to hear. Based on the results presented here, there is a good chance that a number of previously ignored imaging modalities will now have higher value because of what can be done with deep learning. Unfortunately, I don't think the paper as written does a good job of relaying this conceptual shift and this is a substantial issue with the paper. Some of my recommendations to address this would include:

3a) Restructuring the Introduction. The prior work of Greg Johnson and others should be presented earlier so it is clear that this work builds on theirs. Doing so would make it easier for readers to appreciate that the novelty lies in combining these methods with the author's approach to quantitative label-free images.

3b) Better describe the novelty and performance gains. On the label-free imaging perspective, it is unclear how much of the work presented here is novel, as opposed to a straightforward application of the author's previous fluorescence based methods. I think this could be better explained. Also, the advantages of their method with respect to archival samples (i.e., obtaining staining information while avoiding potentially damaging stains) should be described earlier. The benefit of these methods for live-cell imaging (obtaining data while avoiding photodamage with respect to fluorescence) should also be mentioned, albeit with the appropriate reference.

4) In addition to this, the second major issue with this paper is how much of a performance boost does the author's label-free imaging approach provide? While the conceptual shift described above is appealing and should be highlighted, the case the authors make that this transforms one's ability to use image-to-image translation models on biological images is less clear. The authors use both the Pearson correlation and the structural similarity index to quantify their reconstruction of fluorescent actin in U2OS cells and in brain slices. However, the differences between standard label-free imaging (brightfield and phase) and the author's approach (brightfield, phase, retardance, and orientation) appear minor. For instance in Table 2, the difference in Pearson correlation is ~0.01-0.02 (the gap does appear to be bigger for FluoroMyelin, but fewer comparisons are presented). On its surface, this appears to be a minor advance (although one could argue whether it is in the realm of statistical significance) and as an experimentalist, it makes one question whether the "juice" of the author's method is worth the "squeeze". However, there are certainly cases where minor boosts in accuracy lead to a big difference in one's ability to use a method. While the ability to measure orientation is certainly useful for following neural fibers, it feels like the case that this architectural information is critical to infer fluorescence patterns hasn't been made.

---

## [Author Response]

Essential revisions:1) The idea of label-free prediction of a specific tissue structure from density and anisotropy measurements is appealing and well described in the manuscript. However, the selected types of tissue and some of the general conclusions drawn (partially from cross-comparisons) are disputable. What makes mouse kidney, mouse brain and prenatal human brain unique in terms of 'revealing architectural order', but still comparable? There seems to be a lack in knowledge of brain anatomy and morphology, which is important to evaluate the results. Although the GW24 and GW20 measurements are exciting, the shown tiny ROIs are not suitable for highlighting the anatomical differences in a convincing way. The deep learning approaches appear to be correctly implemented and applied, but their scalability does not become obvious (although stated in the Abstract). A critical debate about the efforts needed to address entire large-scale organs is missing. The major add-on to state-of-the-art approaches or to previous own publications does not become clear enough.

In this revision, we name our computational imaging method QLIPP (quantitative label-free imaging with phase and polarization) to clarify how it differs from quantitative phase imaging (QPI), quantitative polarization imaging, or fluorescence polarization imaging methods reported before this paper. We have edited the Abstract and Introduction to identify key advances relative to the state-of-the-art:

a) Joint label-free measurement of density (specimen phase) and anisotropy (retardance and orientation).

b) High-resolution measurements of density and anisotropy of live 3D cells and prenatal human brain tissue section.

c) 2.5D multi-channel convolutional neural network (CNN) to predict fluorescence over large fields-of-view.

Comparisons across data from brain tissue and live cells demonstrate that QLIPP is useful to analyze architecture at multiple spatial scales. The cross-comparisons are meant to, and limited to, illustrate how architecture at multiple scales can be interpreted from the measurements we report. We have re-structured the first result (QLIPP provides joint measurement of specimen density and anisotropy) to emphasize this point.

Mouse kidney tissue is a test specimen without any staining artifacts that we used to develop the architecture of CNN, training methods, and metrics for evaluation of trained models. The data from mouse kidney tissue are now moved to Figures 3 and 4 that report optimization of the CNN architecture.

In the Results and discussions, we’ve clarified questions in the analysis of organelle dynamics and architecture of brain tissue that become tractable with the data we report. Specifically, the videos (Video 1 and 2) of a dividing cell illustrate that organelles can be distinguished by pseudo-coloring the density and retardance values. For mouse brain tissue (Figure 5) and human brain tissue (Figure 6), we visualize architecture over centimeter sized slices. For human brain tissue, we predict myelin images at both Gestational Week 24 (GW24) and GW20 (Figure 7). These data illustrate that quantitative imaging of density and anisotropy, combined with deep learning, can reveal architectural order in diverse brain types.

We agree with the reviewer’s point of view that the scalability of quantitative label-free imaging and deep learning for analysis of the architecture of whole organs needs to be evaluated. The rate limiting factor may not be imaging or analysis, but rather tissue processing. We carefully use the term ‘scale’ only to imply the spatial and temporal scales of measurement and not the ease of using the approach to analyze large-scale organs.

2) Ethics: There is no clear statement of the authors concerning ethical approval, origin of samples, etc. The treatment of prenatal human tissue requires other information than the mouse brain and kidney tissue!

Lack of statement about ethical approval for use of prenatal human tissue was an oversight. We have described the ethical approval in the Materials and methods section.

“De-identified brain tissue samples were received with patient consent in accordance with a protocol approved by the Human Gamete, Embryo, and Stem Cell Research Committee (institutional review board) at the University of California, San Francisco.”

3) The key insight offered by this paper is that because deep learning is data-driven, these methods can be improved by improving data rather than making substantial changes to the algorithms. If there is information missing in the images that is needed to make accurate predictions, why not add it in? To me this is an under-appreciated insight, one that the authors cleverly take advantage of, and one that the life science community as a whole sorely needs to hear. Based on the results presented here, there is a good chance that a number of previously ignored imaging modalities will now have higher value because of what can be done with deep learning. Unfortunately, I don't think the paper as written does a good job of relaying this conceptual shift and this is a substantial issue with the paper. Some of my recommendations to address this would include:

We do agree with the reviewer that informative data is as important as algorithms for data-driven analysis. We also appreciate the reviewers’ inputs on clarity of contributions, which we’ve used to restructure the Introduction and Results as described in response to comment # 1 above.

3a) Restructuring the Introduction. The prior work of Greg Johnson and others should be presented earlier so it is clear that this work builds on theirs. Doing so would make it easier for readers to appreciate that the novelty lies in combining these methods with the author's approach to quantitative label-free images.

We have re-written the Introduction and Discussion to clarify how our work builds upon Greg Johnson’s work by a) adding new modalities of data and b) by adapting their deep learning architecture for computationally efficient training.

3b) Better describe the novelty and performance gains. On the label-free imaging perspective, it is unclear how much of the work presented here is novel, as opposed to a straightforward application of the author's previous fluorescence based methods. I think this could be better explained. Also, the advantages of their method with respect to archival samples (i.e., obtaining staining information while avoiding potentially damaging stains) should be described earlier. The benefit of these methods for live-cell imaging (obtaining data while avoiding photodamage with respect to fluorescence) should also be mentioned, albeit with the appropriate reference.

In terms of label-free imaging, the novel contribution of QLIPP is a more precise forward model of image formation and corresponding inverse algorithms for joint imaging of density and anisotropy using a simple light path. The light-path is identical to the transmission polarized light microscope (Mehta, Shribak and Oldenbourg, 2013), but different from the fluorescence polarization methods reported earlier by the corresponding author (Mehta et al., 2016). We now clearly distinguish QLIPP from other classes of polarization-resolved methods in the Introduction and the Discussion.

4) In addition to this, the second major issue with this paper is how much of a performance boost does the author's label-free imaging approach provide? While the conceptual shift described above is appealing and should be highlighted, the case the authors make that this transforms one's ability to use image-to-image translation models on biological images is less clear. The authors use both the Pearson correlation and the structural similarity index to quantify their reconstruction of fluorescent actin in U2OS cells and in brain slices. However, the differences between standard label-free imaging (brightfield and phase) and the author's approach (brightfield, phase, retardance, and orientation) appear minor. For instance in Table 2, the difference in Pearson correlation is ~0.01-0.02 (the gap does appear to be bigger for FluoroMyelin, but fewer comparisons are presented). On its surface, this appears to be a minor advance (although one could argue whether it is in the realm of statistical significance) and as an experimentalist, it makes one question whether the "juice" of the author's method is worth the "squeeze". However, there are certainly cases where minor boosts in accuracy lead to a big difference in one's ability to use a method. While the ability to measure orientation is certainly useful for following neural fibers, it feels like the case that this architectural information is critical to infer fluorescence patterns hasn't been made.

To clarify the performance boost achievable using anisotropy data (retardance and orientation), we have added Figure 7 and updated Table 4. We report predicted images of FluoroMyelin over large fields of view and analyze the accuracy of prediction as a function of input data: only bright field; retardance and phase; retardance, phase, and orientation; brightfield, phase, and orientation. As compared to conventionally available data (brightfield), use of QLIPP data provides a significant increase in the accuracy of prediction as can be seen from images in Figure 7B and Figure 7E. We have added plots (Figure 7C, Figure 7F, and Figure 7—figure supplement 1) that show correlation of FluoroMyelin target with input retardance, phase, and FluoroMyelin predicted by our models. As shown in Table 4, using QLIPP data leads to an increase of 0.14 for both Pearson correlation and SSIM between prediction and target, relative to using just brightfield data. The prediction is more accurate with QLIPP data, because the pattern of myelination in the brain is encoded in anisotropy. Although this increase is small on the scale of these metrics (14%), it is biologically significant as can be seen from the images.

One known issue with image similarity metrics such as Pearson correlation and SSIM is that they can be insensitive to small, but biological relevant features in the image. This is because Pearson correlation and SSIM report the average similarity over the image with every pixel equally weighted.

https://ieeexplore.ieee.org/document/5705575;

https://arxiv.org/abs/1406.7799;

https://ieeexplore.ieee.org/abstract/document/5596999

We illustrate above point using Mouse Kidney tissue data (Figure 4—figure supplement 1). The image similarity metrics can change by the same small value when white noise is added or when a nucleus is removed. These sources of distortions cannot be distinguished with image similarity metrics as they do not fully exploit the spatial context. We have also revised the text in the Results (Optimization of 2.5D model with a test dataset) to emphasize this point.

A more informative metric would weigh pixels with biological information more than other pixels, e.g. a missing nucleus should be weighted more than mismatch in the background fluorescence. We considered image quality metrics that mimic human perception of image quality, e.g. multi-scale SSIM, feature similarity index, and CNN based predictors of quality. However, their applicability to quantitative evaluation of similarity between biological image sets is yet to be established. We chose to report Pearson correlation and SSIM because they are more commonly used in the biology and imaging community. Finding a biologically relevant metric of accuracy of image prediction is an interesting future research topic.